# Revisiting Long-context Modeling from Context Denoising Perspective

**Zecheng Tang**[1,2], **Baibei Ji**[1,2], **Juntao Li**[1,2*], **Lijun Wu**[3], **Haijia Gui**[1], **Min Zhang**[1]

[1]Soochow University  [2]LCM Laboratory  [3]Shanghai Artificial Intelligence Laboratory

{zctang, bbji}@stu.suda.edu.cn  {ljt, minzhang}@suda.edu.cn

## Abstract

Long-context models (LCMs) have demonstrated great potential in processing long sequences, facilitating many real-world applications. The success of LCMs can be attributed to their ability to locate implicit critical information within the context for further prediction. However, recent research reveals that LCMs are often susceptible to contextual noise, i.e., irrelevant tokens, that can mislead model attention. In this paper, we conduct a fine-grained analysis of the context noise and propose an effective metric, the Integrated Gradient (IG) score, to detect and quantify the noise information within the context. Our findings reveal that even simple mitigation of detected context noise can substantially boost the model's attention on critical tokens and benefit subsequent predictions. Building on this insight, we propose Context Denoising Training (CDT), a straightforward yet effective training strategy that improves attention on critical tokens while reinforcing their influence on model predictions. Extensive experiments across four tasks, under both context window scaling and long-context alignment settings, demonstrate the superiority of CDT. Notably, when trained with CDT, an open-source 8B model can achieve performance (50.92) comparable to GPT-4o (51.00). [1].

## 1 Introduction

The ability to handle long input sequences has become a fundamental requirement for large language models (LLMs), with cutting-edge models capable of processing context lengths exceeding millions of tokens (Team et al., 2024; MiniMax et al., 2025; Meta, 2025; Qiu et al., 2025b). This advancement eliminates the need for complex toolchains and intricate workflows, e.g., RAG (Yu et al., 2024), and significantly enhances real-world applications, such as LLM agent (Luo et al., 2025; Xi et al., 2025) and project code analysis (Fang et al., 2024a).

Recent studies indicate that LCMs frequently fail when processing long-context tasks (Hsieh et al., 2024; Kuratov et al., 2024; Tang et al., 2024b; Bai et al., 2024c), and the open-source community mitigates such an issue mainly by using sufficient high-quality synthetic long-context data to post-train the model (Fu et al.,

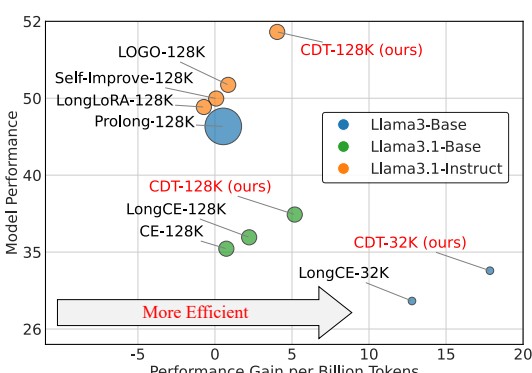

Figure 1: Comparative overview of *model performance* on real-world long-context tasks and *performance gain per billion tokens* among different training methods. The bubble size indicates the relative training data volume.

2024a; Chen et al., 2024b; Gao et al., 2024a). However, these approaches are proven to be either inefficient or ineffective under limited resources. A controlled study in Appendix A shows that, when trained on 2B tokens using the Llama3-8B backbone, Prolong-64K-Base (Gao et al., 2024b) improves its average score on 12 real-world tasks from 25.5 to 29.13-equivalent to a gain of 1.8

---

*Corresponding Author

[1] ⌂ Code is available at: https://github.com/LCM-Lab/context-denoising-training

points per additional 1B tokens—whereas LongCE (Fang et al., 2024b) achieves 32.91 points, corresponding to a gain of 3.7 points per 1B tokens. As illustrated in Figure 1, the training efficiency decreases as the training context length increases to 128K, e.g., Prolong-128K.

One of the possible reasons is that existing works overlook the fact that LCMs process long input in an implicit *retrieval-then-generation* manner, i.e., first identifying key information within the context and then further generating with the "retrieved-context" (Liu et al., 2024b; Wu et al., 2024; Li et al., 2024a; Qiu et al., 2025a). However, the critical tokens in the "retrieved-context" might be overwhelmed by excessive irrelevant tokens (Ye et al., 2024). Thus, the key to achieving better long-context modeling is *effectively detecting the critical tokens, diminishing the effect of irrelevant tokens (context noise), and strengthening the connection between model prediction and critical tokens.* Conventional language modeling training strategy, which relies on uniform token-wise supervision through cross-entropy loss, is fundamentally inefficient for long-context modeling because it cannot distinguish critical tokens from irrelevant tokens in lengthy inputs.

In this work, we first investigate the impact of context noise on long-context modeling. Specifically, we propose a novel critical token detection metric, the Integrated Gradient (IG) score, based on the concept of information flow (Wang et al., 2023). Our approach achieves a remarkable accuracy improvement in the critical token detection task compared to the traditional attention-based method. Then, we leverage the IG score to manually reduce the context noise by subtracting the gradient values associated with irrelevant tokens from the token embeddings. We find that simply suppressing context noise at the model input allows LCMs to focus more effectively on critical tokens.

Built upon the above analysis, we further propose a simple yet effective Context Denoising Training (CDT) strategy, which performs denoising at the model input, allowing the model to focus more effectively on critical tokens to better establish the connection between critical tokens and generation. To adapt CDT to long-sequence training and further improve training efficiency while reducing peak memory consumption, we theoretically derive a method that leverages gradients with respect to token embeddings (Appendix C)—rather than directly using IG scores mentioned above—as identifiers to detect noisy tokens. Notably, our CDT approach is analogous to the *Signal Denoising* in the digital signal processing field (Kopsinis & McLaughlin, 2009), where noise reduction in the input sequence can enhance the model's attention to essential parts within the context. Experiments on two essential long-context training scenarios, i.e., context window scaling and long-context alignments, across 4 different types of long-context tasks (real-world tasks, language modeling task, synthetic tasks, and long-form reasoning tasks) exhibit the superiority of our method. Our CDT can consistently surpass the other methods with an average gain of 2 points on 12 real-world long-context tasks in LongBench-E Bai et al. (2024b) and 13 long synthetic tasks in RULER (Hsieh et al., 2024). Additionally, with CDT, an open-source Llama3.1-8B-Instruct model can achieve comparable results with GPT4o on real-world tasks (50.92 points v.s. 51.00 points on LongBench-E testing set).

## 2 RELATED WORK

### 2.1 LONG-CONTEXT POST-TRAINING

Generally, the purposes of long-context post-training can be categorized into two types: *context window scaling* and *long-context alignment*. For context window scaling, prior studies have managed to extend the context length of LLMs with limited computational cost compared to pretraining. It can be further categorized into two approaches: positional extrapolation(Chen et al., 2023a; Peng et al., 2023; Ding et al., 2024; Liu et al., 2024a; Zhao et al., 2024a; Zhang et al., 2024c; Fu et al., 2024b; Lu et al., 2024; Wang et al., 2025b; Ge et al., 2025; Xiong et al., 2024; Chen et al., 2024a; Liu et al., 2024c) and model architecture modification(Chevalier et al., 2023; Chen et al., 2023b; Xiao et al., 2024b; Bertsch et al., 2024; Yuan et al., 2025; Lu et al., 2025). Another line of work focuses on improving models that already support long-context windows, aiming to enhance the model's ability to capture critical information from lengthy contexts (Liu et al., 2024b; An et al., 2024b; Gao et al., 2024c; An et al., 2024a) and to address alignment challenges such as hallucination (Zhang et al., 2024b; Tang et al., 2024a; Li et al., 2024b). However, to date, no existing work has simultaneously considered both training efficiency and effectiveness under the two aforementioned settings. Only a few preliminary studies (Lin et al., 2024; Fang et al., 2024b; Helm et al., 2025; Wang et al., 2025a) have explored token re-weighting as a trivial method to achieve a limited trade-off.

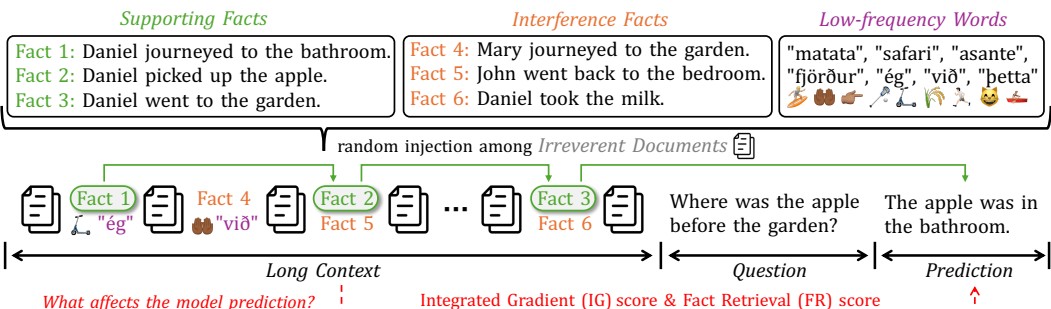

Figure 2: Task format of our preliminary study, which requires models to predict the final answer by reasoning through multi-hop Supporting Facts and distinguishing from the Interference Facts. Simultaneously, the model should also resist the influence of Irreverent Documents and Low-Frequency Words. More details are shown in Appendix B.

## 2.2 RETRIEVAL-THEN-GENERATION MECHANISM OF LONG-CONTEXT MODELS

Existing research has demonstrated that LCMs handle long-context in a "retrieval-then-generation" manner, where *LCMs first retrieve salient information within the context and utilize this information for further prediction* (Wu et al., 2024; Tang et al., 2024b; Zhao et al., 2024b; Qiu et al., 2025a). However, Liu et al. (2024b) observes the "lost-in-the-middle" phenomenon of LCMs, which highlights that LCMs exhibit a positional bias toward locating key information. Furthermore, Ye et al. (2024) and Fang et al. (2024b) discover that excessive irrelevant long-context can overwhelm critical information, thereby impairing the performance of the model. To mitigate the above issue, some works have explored solutions from various perspectives, including model architecture improvements (Ye et al., 2024; Xiao et al., 2024a), enhancements in information extraction mechanisms (Li et al., 2024a; Zhang et al., 2024a), and optimization of training objective (Fang et al., 2024b; Bai et al., 2024a). In this paper, we revisit critical information location from the context denoising aspect, helping the model establish better connections between detected salient tokens and predictions.

## 3 PRELIMINARY STUDY

In this section, we analyze the influence of context noise, i.e., irrelevant tokens, on long-context modeling. More concretely, we first design critical token detection metrics in §3.1 and study the impact of context noise restraint on long-context modeling in §3.2. For evaluation, we construct a synthetic long-form reasoning task as a controlled proxy to enable precise assessment, due to the lack of real-world testing data with explicitly labeled critical token positions. We conduct experiments with the Llama3.1-8B-Instruct (Meta, 2024) model, which owns a 128K context window size.

**Synthetic Task Format**  As shown in Figure 2, there are four types of tokens in the context: supporting facts, interference facts, low-frequency words, and irrelevant documents. The model's task is to predict the correct answer (e.g., "bathroom") by reasoning over supporting facts. The interference facts are seemingly related to the answers and are randomly inserted into the context, aiming to distract the models from providing the correct response. We treat both supporting facts and interference facts as *critical tokens*, as they are both highly correlated with the answer. The key distinction lies in semantic validity: LCMs must discern which tokens are genuinely supportive — and which are misleading — to predict accurately. Besides, models should also prevent critical tokens from being overwhelmed by *irrelevant tokens*, including excessive irrelevant documents and low-frequency words. The total context length of each sample ranges from 0K to 64K.

## 3.1 CRITICAL TOKENS DETECTION

Given the model input $X = \{x_i\}_{i=1}^{n}$ which contains $n$ tokens and the ground truth $Y = \{y_j\}_{j=1}^{m}$ which contains $m$ tokens, we design two metrics to reflect the influence of context noise: Fact Retrieval (FR) score and Integrated Gradient (IG) score.

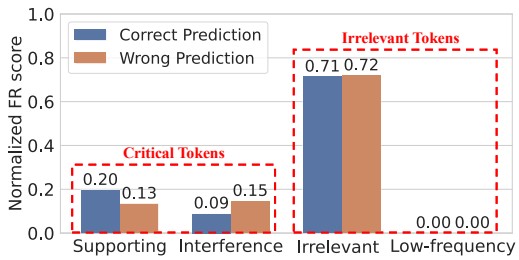
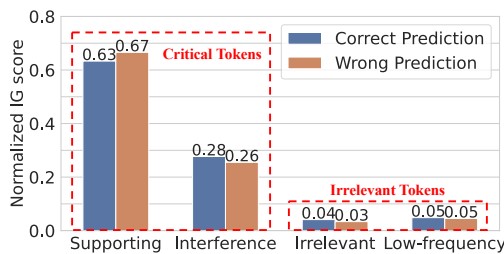

(a) Attention distribution reflected by FR score.

(b) Information flow reflected by average IG score.

Figure 3: Comparison between attention distribution and information flow on the critical token location task. A significant difference in the distributions of critical and irrelevant contexts is revealed.

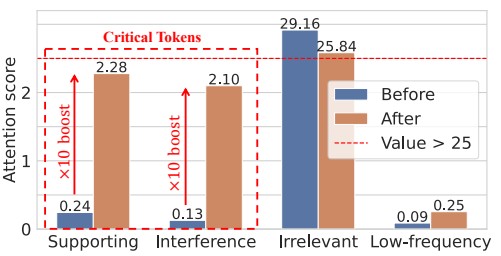

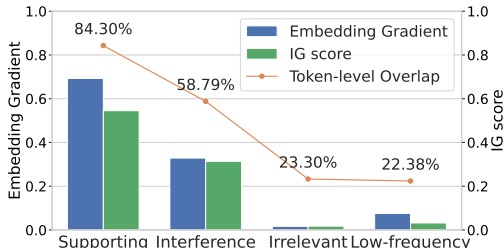

Figure 4: Attention distributions before and after manual context denoising. After context denoising, attention scores on critical tokens boost $\times 10$ times, and show a reduction on irrelevant tokens.

Figure 5: Relationship between attention IG score and L2-normalized embedding gradients on different types of tokens. It shows a proportional correlation.

**Attention Distribution Metric: FR score**   Existing works primarily identify critical tokens based on the attention distribution (Wu et al., 2024; Gema et al., 2024; Xiao et al., 2024a). Similarly, we design the Fact Retrieval (FR) score for our synthetic task based on the attention distribution to quantify the model's attention allocated to different types of tokens. At each step of model prediction $y_j$, if the attention score of $x_i$ ranks within the top-k across the entire sequence, we define $x_i$ as being attended by an attention head $h$ in the $l$-th model layer. Let $s_j$ be the set of tokens attended by an attention head $h$ at the generation step $j$, and $\mathcal{T}_r$ refers to the context token set of type $r \in \{\text{sup}, \text{inter}, \text{irr}, \text{low}\}$, e.g., $\mathcal{T}_{sup}$ denotes tokens of the supporting facts. The FR score $\text{FR}_{h,l}^{(r)}$ of the $h$-th attention head in the $l$-th model layer can be written as:

$$\text{FR}_{h,l}^{(r)} = \frac{\mid s_j \cap \mathcal{T}_r \mid}{\mid \mathcal{T}_r \mid}.$$

We average FR scores from all heads to reflect the attention distribution of tokens in $\mathcal{T}_r$.

**Information Flow Metric: IG score**   To discover the attention interaction among tokens, i.e., information flow (Simonyan et al., 2013), we employ the Integrated Gradient (IG) technique (Wang et al., 2023). We define the IG score of $h$-th head in model's $l$-th layer on segment $\mathcal{T}_r$ below:

$$\text{IG}_{h,l} = A_{h,l}^T \odot \mid \frac{\partial \mathcal{L}_\theta(Y|X)}{\partial A_{h,l}} \mid, \quad \text{IG}_{h,l}^{(r)} = \frac{1}{|\mathcal{T}_r|} \sum_{x_i \in \mathcal{T}_r} \sum_{y_j \in Y} \text{IG}_{h,l}[i,j], \tag{1}$$

where $\mathcal{L}_\theta(Y|X)$ is the model's prediction loss on $Y$, and $A_{h,l}$ denotes the attention matrix of the $h$-th head in the $l$-th layer. The resulting IG score is a matrix, where each entry $\text{IG}_{h,l}[i,j]$ represents the estimated bidirectional information flow between token $x_i$ and token $y_j$. To assess the overall impact of $\mathcal{T}_r$ to $Y$, we compute the total contribution of tokens in $\mathcal{T}_r$ to the final prediction $Y$, i.e., $\text{IG}_{h,l}^{(r)}$ and average across all attention heads and layers as the final score, i.e., $\text{IG}^{(r)}$. A higher IG score $\text{IG}^{(r)}$ indicates a larger contribution from $\mathcal{T}_r$ to $Y$. Details are shown in Appendix B.2.

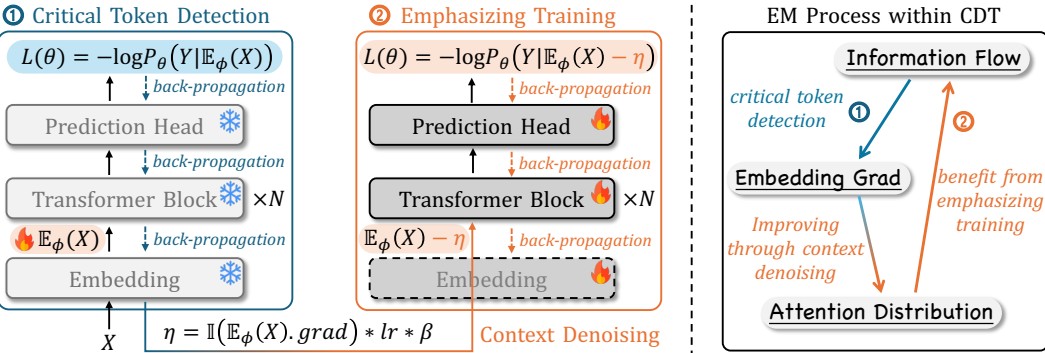

Figure 6: Our proposed CDT (context denoising training) method. It consists of two steps: (1) detecting critical tokens within the long context, and (2) utilizing the denoised context for further emphasizing training. Notably, CDT can be understood as an *Expectation Maximization (EM)* process, where the model detects noise based on information flow and improves the training by diminishing the noise, thereby enhancing the information flow.

**Observation** For a clear comparison, we normalize the computed FR and IG scores, and plot them in Figure 3. We find that the IG score detects significantly less noise (irrelevant documents and low-frequency tokens) compared to the FR score on critical token detection. Specifically, as shown in Figure 3a, attention-based metrics reflect the distribution of tokens that the model focuses on during the generation process. When the model generates correct responses, its attention focuses more on supporting facts; when the model generates wrong responses, its attention focuses more on interference tokens. *Yet, in both cases, the FR score indicates that the model significantly focuses on irrelevant tokens.* As for the IG score shown in Figure 3b, regardless of whether the response is correct or not, *the IG score for critical tokens is significantly higher than that for irrelevant tokens.*

## 3.2 EFFECT OF MANUAL CONTEXT NOISE RESTRAINT

Considering that directly suppressing context noise in attention is very challenging, we aim to restrain the noise from the input perspective. We first identify irrelevant tokens by computing the IG score on each token and treating the token with the IG score lower than a threshold as the noisy token. Then, we manually suppressed their influence by subtracting the corresponding gradients from their input embeddings. This is motivated by the fact that the model has largely converged on these noisy tokens, resulting in their gradients exhibiting low sensitivity. As shown in Figure 4, we observe that after manual context denoising, the attention scores on critical tokens increase nearly $\times 10$ times, while the attention scores on irrelevant contextual tokens exhibit a slight decrease. It is worth noting that this operation can be roughly analogized to *denoising in the digital signal processing field* (Kopsinis & McLaughlin, 2009), as it reduces noise in the input sequence, allowing the model to focus more effectively on the under-fitting critical tokens.

## 4 CONTEXT DENOISING TRAINING

Based on the above observation, we propose a simple yet effective Context Denoising Training (**CDT**) strategy. Building upon the conventional language modeling objective, i.e., cross-entropy loss, CDT explicitly suppresses context noise during training to strengthen the model's attention on critical tokens and help establish a better connection between critical tokens and the final prediction. It involves two key steps: (1) Critical Token Detection and (2) Emphasizing Training.

### 4.1 CRITICAL TOKEN DETECTION

Intuitively, we can first apply IG score to detect the critical tokens for the subsequent training. However, computing the IG score in long-context scenario is highly GPU memory-intensive, as it requires storing full attention gradients and weights from every model layer across the entire sequence. **Even with 8×92GB GPUs (H20), the maximum computable sequence length for the IG score is limited to 12K**, making it infeasible to generalize to a longer sequence. Therefore, we

designed a simple alternative implementation, which approximates the IG score with token embedding gradients[2]. We derive a proportional relationship between the token embedding gradient and the IG score, and visualize the results in Figure 5. A detailed derivation is provided in Appendix C.

As shown in Figure 6, given the input sequence $X = \{x_i\}_{i=1}^n$, label $Y$, and the model $f_\theta$, we first freeze the model parameters, keeping only the gradients of the input token embeddings $E_\phi(X)$, where $\phi \subset \theta$. We then obtain the gradient of each token embedding through the computation of the cross-entropy (CE) loss followed by a loss back-propagation. To identify the critical tokens, we employ an identifier $\mathbb{I}(\cdot)$ to detect tokens with large gradients, i.e., critical tokens, in the sequence. Specifically, we define the calculation of the significance of each token as comparing its L2-normalized embedding gradient against the average of the computed gradients of all tokens, which can be written as:

$$\mathbb{I}(x_i) = \begin{cases} 1, & \text{if } ||\nabla_{E_\phi(x_i)}\mathcal{L}_{\text{CE}}(x_i)||_2 < t \\ 0, & \text{if } ||\nabla_{E_\phi(x_i)}\mathcal{L}_{\text{CE}}(x_i)||_2 \geq t \end{cases}, \quad t = \frac{1}{n}\sum_{i=1}^n ||\nabla_{E_\phi(x_i)}\mathcal{L}_{\text{CE}}(x_i)||_2, \quad (2)$$

where $\mathbb{I}(x_i) = 1$ means $x_i$ is the irrelevant token (noise); otherwise, it is critical token.

## 4.2 EMPHASIZING TRAINING

To suppress the context noise, we leverage the computed gradients to manipulate the irrelevant token embeddings, leaving critical tokens unchanged. The denoised token embedding can be denoised as:

$$E_\phi(x_i)' = E_\phi(x_i) - \mathbb{I}(x_i)\nabla_{E_\phi(x_i)} \times lr \times \beta, \quad (3)$$

where $lr$ is the learning rate and $\beta$ is the hyper-parameter controlling the denoising level. Then, we unfreeze the model and use the denoised token embeddings as the model input for further training, which we refer to as Emphasizing Training. The loss function can be formulated as:

$$\mathcal{L}_{CDT}(X, Y) = \mathcal{L}_{CE}\left(f_\theta\left(E_\phi(X)'\right), Y\right). \quad (4)$$

**Remark** Notably, the above process is conducted online during training rather than pre-computed offline. As shown in Figure 6, although this introduces additional computational overhead, CDT bootstraps the model's long-context capabilities in an *Expectation-Maximization (EM) manner*: the model first identifies the critical tokens based on information flow and improves the training by diminishing the noise, thereby ultimately enhancing the information flow. In § 6.3, we will demonstrate that, by training with CDT, the model can continuously enhance its performance compared to conventional training objectives during the post-training stage.

## 5 EXPERIMENT

### 5.1 EXPERIMENTAL SETUPS

**Evaluation** We evaluate models on 4 different types of long-context tasks, including real-world tasks (LongBench-E (Bai et al., 2024b), language modeling task (LongPPL (Fang et al., 2024b)), long-form reasoning task (BABILong (Kuratov et al., 2024)), and synthetic tasks (RULER (Hsieh et al., 2024)). We compare CDT against existing widely-used methods on two types of models: (1) short-context models (SCMs) that require context window scaling; (2) long-context models (LCMs) that require long-context alignment. In our main experiments, we select Llama-3-8B-Base model as the SCM, of which context window size is scaled $\times 8$ times (64K). For LCMs, we select Llama-3.1-8B-Base and Llama-3.1-8B-Instruct models. We provide more evaluation and baseline details in Appendix D, and show more evaluation results, such as generalizing CDT to more models,e.g., Qwen-series (Yang et al., 2024; 2025)), in Appendix E. We evaluate against current strong LCMs, as well as diverse long-context enhancement methods across training and inference paradigms — including token-wise reweighting (LongCE (Fang et al., 2024b)), KV-cache prefilling (Lai et al., 2025), SFT (Chen et al., 2024b), and RL-based optimization (Tang et al., 2024a).

---

[2]We choose token embeddings for 3 reasons: (1) they are easily accessible, (2) the embedding gradients are directly associated with tokens, and (3) they require much less GPU memory compared to attention gradients.

Table 1: Evaluation results on LongBench-E benchmark. To ensure fairness, we place existing works that do not use the same training data with us in the top group. Our method is implemeted under three settings: context-window scaling (CWS), language modeling (LM), and SFT.

| Models | Type | S-Doc QA | M-Doc QA | Summ | Few-shot | Code | Avg. |
|---|---|---|---|---|---|---|---|
| ProLong-512K-Instruct (Gao et al., 2024b) | SFT | 40.07 | 41.24 | 28.27 | 64.21 | 63.08 | 47.37 |
| NExtLong-512K-Instruct (Gao et al., 2025) | SFT | 43.47 | 43.21 | 29.88 | 60.87 | 44.35 | 44.35 |
| Llama-3.1-8B-SEALONG (Li et al., 2024b) | DPO | 49.45 | 44.69 | **30.96** | 61.54 | 57.85 | 48.90 |
| GPT-4o (version: 2024-11-20) | - | **51.43** | **60.89** | 14.78 | **66.37** | **61.25** | **51.00** |
| **Results on Short-context Model** (*all SCMs share the same training data, $8\times$ context window scaling.*) | | | | | | | |
| Llama-3-8B-Base (8K) | - | 25.20 | 21.52 | 20.18 | 32.67 | 27.92 | 25.50 |
| + YaRN (Peng et al., 2023) | - | 24.37 | 19.86 | 24.32 | 29.99 | 31.67 | 26.04 |
| + CE | CWS | 25.29 | 21.49 | 20.36 | 32.69 | 27.76 | 34.62 |
| + LongCE (Fang et al., 2024b) | CWS | 17.13 | 9.59 | 25.00 | 59.57 | 61.83 | 34.62 |
| + CDT (ours) | CWS | 17.03 | **24.87** | **26.61** | 61.89 | 66.14 | **39.31** |
| **Results on Long-context Base Model** (*all LCMs share the same training data.*) | | | | | | | |
| Llama-3.1-8B-Base | - | 18.20 | **13.19** | 21.13 | **63.80** | 69.32 | 37.13 |
| + CE | LM | 17.10 | 10.82 | 26.38 | 62.85 | **70.62** | 37.55 |
| + LongCE (Fang et al., 2024b) | LM | 19.14 | 10.87 | 28.63 | 59.63 | 66.24 | 36.90 |
| + CDT (ours) | LM | **19.15** | 13.01 | **29.23** | 63.63 | 69.44 | **38.89** |
| **Results on Long-context Instruct Model** (*all LCMs use same source data with different formats.*) | | | | | | | |
| Llama-3.1-8B-Instruct | - | 48.58 | 45.19 | 30.30 | 61.73 | 57.26 | 48.61 |
| + Contriever (Izacard et al., 2021) | RAG | 42.63 | 45.55 | 32.48 | 62.15 | 41.85 | 44.93 |
| + FlexPrefill (Lai et al., 2025) | KV-Prefill | 47.02 | 45.55 | 27.37 | 60.97 | 55.97 | 47.38 |
| + X-Attention (Xu et al., 2025) | KV-Prefill | 48.32 | 45.60 | 26.93 | 61.83 | 56.39 | 47.81 |
| + SFT | SFT | 49.23 | 44.86 | 30.39 | 61.96 | 57.14 | 48.72 |
| + LOGO (Tang et al., 2024a) | DPO | 49.63 | 45.39 | **30.44** | 62.39 | 57.19 | 49.01 |
| + CDT (ours) | SFT | **50.11** | **46.04** | 30.34 | **62.49** | **65.64** | **50.92** |

**Training and Datasets** For context window scaling training on SCM and post-training on LCM-Base, we apply PG-19 (Rae et al., 2019) as the training data. For each training sample, we organize it into 64K tokens and collect 10,000 training samples. For long-context alignment on LCM-Instruct, we utilize data sampled from LongMiT (Chen et al., 2024b) and LongAlpaca (Chen et al., 2023c), covering 8,000 samples with context lengths ranging from 16K to 128K. Based on the analysis experiment (Section 6.2), we set $\beta = 5$ in Equation 3 for Llama-3.1 and Llama-3 models in the main experiments. More dataset processing and implementation details are shown in Appendix D

## 5.2 RESULTS

**Real-world Long-context Understanding Tasks** LongBench-E is a comprehensive benchmark suite encompassing 12 real-world datasets and various context lengths spread across 5 categories. As shown in table 1, we observe that: **(1)** *CDT achieves the best performance among all the sub-tasks*. For SCMs, with the same training data, CDT achieved the best performance, outperforming a competitive counterpart (LongCE) by nearly 4.7 points on average. **(2)** For LCM-Base models, we find when post-training on the base model with language modeling training objective, *CDT is the only method that ensures no significant performance drop across all subtasks*, and it even achieves some improvements. In contrast, using standard CE or LongCE objective leads to significant performance drops on some sub-tasks. For example, LongCE results in a nearly 4-point drop compared to the backbone model on the Few-shot subtask. **(3)** As for the LCM-Instruct models (the bottom group), we find that, due to its remarkable performance, *existing post-training methods do not bring significant improvements*. For instance, Llama-3.1-8B-SEALONG (48.90) achieves only around slight 0.3-point average improvement compared to Llama-3.1-8B-Instruct (49.61). However, our CDT achieves an average improvement of more than 2 points compared to that of the backbone model across all tasks. We provide more analysis of results in Appendix E.2.

**Long Synthetic Task and Language Modeling** For the long synthetic task, we evaluate the model's performance under 32K, 64K, and 128K context lengths. We choose 13 sub-tasks from the RULER benchmark and report the average results. For the language modeling task, we calculate LongPPL (Fang et al., 2024b) on the GovReport dataset (Huang et al., 2021). Notably, LongPPL can potentially reflect the model's ability to locate salient tokens in the long context. More imple-

Table 2: Evaluation results on long synthetic tasks (RULER), language modeling, and long-form reasoning (BABILong). For RULER, we report the average scores across 13 sampled sub-tasks. To calculate LongPPL, we apply the Llama3-8B-Base model as the evaluator. For BABILong, we report the model reasoning capability from short context (4K) to long context (128K).

| Models | RULER | | | Language Modeling | BABILong | | | | | | |
|---|---|---|---|---|---|---|---|---|---|---|---|
| | 32K | 64K | 128K | LongPPL | 4K | 8K | 16K | 32K | 64K | 128K | Avg. |
| ProLong-512K-Instruct | 91.68 | 87.53 | 80.03 | 2.97 | 44.00 | 45.40 | 39.20 | 35.00 | 35.00 | 29.80 | 36.88 |
| NExtLong-512K-Instruct | 90.27 | 84.62 | 81.74 | 3.24 | 39.60 | 38.60 | 36.20 | 35.60 | 32.00 | 22.00 | 38.75 |
| Llama-3.1-8B-SEALONG | 91.32 | 85.97 | 77.33 | 3.09 | 50.20 | 50.80 | 42.00 | 40.80 | 39.00 | 31.00 | 40.72 |
| Llama-3-8B-Base | - | - | - | > 100 | 33.40 | 26.60 | 4.80 | 0.00 | 0.20 | - | 13.00 |
| + YaRN | 39.58 | 31.46 | - | 3.55 | 35.20 | 29.80 | 24.40 | 20.20 | 17.60 | - | 25.44 |
| + CE | 36.01 | 13.82 | - | 3.90 | 36.60 | 34.80 | 26.60 | 28.20 | 21.60 | - | 29.56 |
| + LongCE | 84.02 | 71.50 | - | 3.55 | 36.00 | **34.80** | 34.60 | **32.60** | 29.40 | - | 33.48 |
| + CDT (ours) | **84.76** | **73.40** | - | **3.04** | **38.40** | 34.60 | **34.80** | 31.40 | **29.60** | - | **33.76** |
| Llama-3.1-8B-Base | 89.99 | 81.96 | 70.60 | 3.22 | 35.00 | 33.20 | 27.80 | 28.00 | 25.20 | 24.40 | 28.93 |
| + CE | 86.59 | 80.87 | 70.44 | 3.28 | **39.20** | 31.60 | 31.40 | 26.60 | 26.80 | 19.40 | 29.17 |
| + LongCE | 87.65 | 81.79 | 70.79 | 3.24 | 37.80 | 33.40 | **33.60** | **32.60** | 27.60 | 24.60 | 31.60 |
| + CDT (ours) | **90.36** | **82.23** | **74.12** | **2.10** | 38.80 | **36.60** | 33.20 | 29.40 | **28.20** | **28.20** | **32.40** |
| Llama-3.1-8B-Instruct | 92.49 | 85.98 | 76.71 | 4.05 | 46.60 | 49.60 | 42.40 | 38.80 | 37.00 | 29.60 | 40.67 |
| + SFT | 92.49 | 86.22 | 77.33 | 3.31 | 47.00 | 49.40 | **43.60** | 41.20 | 37.40 | 30.40 | 41.50 |
| + LOGO | 92.54 | 86.93 | 77.68 | 4.11 | 48.20 | 50.00 | 42.60 | 42.20 | 37.40 | 31.60 | 42.00 |
| + CDT (ours) | **93.08** | **88.01** | **78.72** | **2.36** | **51.40** | **51.20** | 41.60 | **44.00** | **38.60** | **33.00** | **43.30** |

mentation and calculation details are illustrated in Appendix D.2. We show the evaluation results in Table 2,where our CDT method achieves the best model performance on the RULER benchmark from 32K to 128K settings. Besides, in the language model task, CDT exhibits the lowest LongPPL, indicating the great potential of CDT to locate salient tokens.

**Short-context & Long-form Reasoning Tasks**  We evaluate the model's long-form reasoning capabilities, as well as its short-context capability, on BABILong, a synthetic task that requires models to reason through multiple supporting facts hidden in contexts of varying lengths (from 4K to 64K). As shown in Table 2, our CDT achieves the highest overall score in each group. Besides, we observe that our CDT does not compromise the model's performance on short-context tasks. For instance, in the 4K and 8K lengths, CDT achieves either the best or comparable results compared to other methods and backbone models.

## 6 ABLATION STUDY

In this section, we compare the accuracy of salient token detection of CDT with other detection methods in §6.1. Then, we show the impact of context denoising on the training process in §6.2. Finally, we elaborate on the training budget of our CDT method in §6.3. Notably, to help better understand the effectiveness of our CDT method, we also analyze the attention map patterns to reveal how CDT influences the model's attention distribution in Appendix F.

### 6.1 COMPARISON OF CRITICAL TOKEN DETECTION

We compare three different detection methods, including LongPPL, attention-based detection, and our CDT, on our synthetic task (Figure 2). For attention-based and our CDT methods, we treat the tokens with the top-30 highest attention scores and L2 normalized gradient of embedding as the detected tokens. As shown in Figure 7, we can observe that the attention-based method can detect a high proportion of supporting tokens and interference tokens, but it also detects a large number of irrelevant tokens. On the other hand, while LongPPL can effectively suppress the detection of irrelevant tokens, it struggles to locate supporting tokens. Our CDT method not only identifies the largest number of critical tokens but also effectively suppresses the detection of irrelevant tokens.

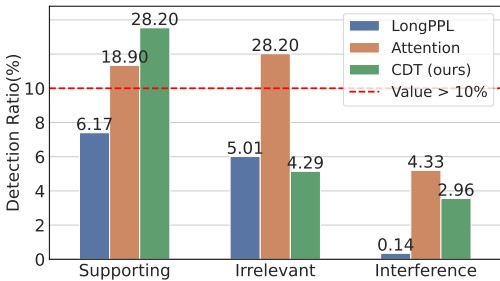

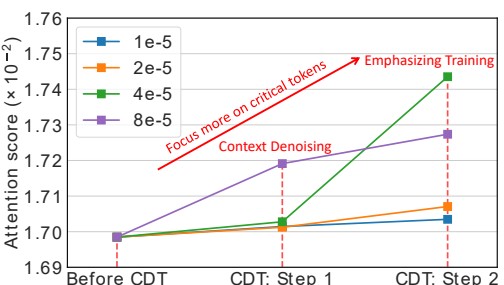

Figure 7: Comparison of critical token detection capability among different methods on our synthetic task. CDT achieves best performance.

Figure 8: Impact of context denoising and comparison of the effect of learning rate on attention scores assigned to critical tokens in CDT.

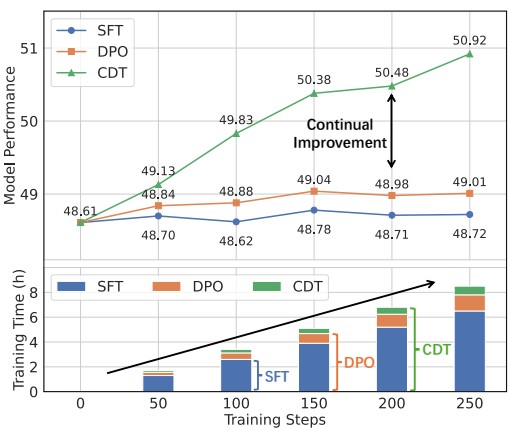

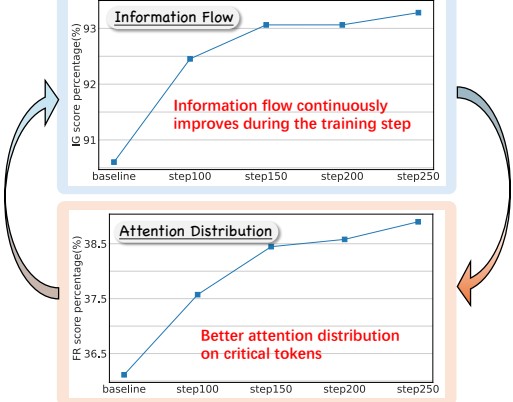

Figure 9: The performance improvement and training duration for every interval of 50 steps. With only a modest cost in training time, CDT significantly boosts the performance of LCM.

Figure 10: Illustration of *EM* process of our CDT method, where both the information flow and attention distribution progressively improve within the training steps.

## 6.2 IMPACT OF CONTEXT DENOISING STRENGTH

We visualize the changes in attention scores allocated to critical tokens during the CDT training process under different learning rates and the same $\beta = 1$ settings. As shown in Figure 8, we observe that the attention scores on critical tokens have already increased significantly after the context denoising step. Furthermore, after the Emphasizing Training stage, there is an additional improvement. Additionally, we observe that a larger learning rate results in more pronounced improvements, further enhancing context denoising. However, a saturation point exists (e.g., at 8e-5), beyond which the benefits plateau. Based on this observation, we adopt a learning rate $lr$ of 1e-5 and set $\beta = 5$ in our main experiments, where $lr \times \beta = 5e - 5$. We also recommend viewing the attention map provided in Appendix F, which shows that CDT enables the model to focus more on key information within long context, without substantially changing the original attention distribution.

## 6.3 TRAINING BUDGETS AND *EM* PROCESS

Compared to conventional long-context training, which performs one forward and one backward pass to update all parameters, CDT introduces an additional noise detection step. Critically, in long-context training, backward passes are typically 2–3× slower than forward passes due to activation recomputation (Shoeybi et al., 2019). Yet CDT adds merely one lightweight backward (where the vast majority of model parameters are frozen) and one extra forward, resulting in minimal wall-clock overhead relative to standard training. We compare CDT with SFT (single Forward-Backward) and DPO (one batch contains pairwise samples) methods. As shown in Figure 9, we observe that although CDT brings additional cost, i.e., approximately 0.5 hours in 8×A100 GPUs for every 50 steps compared with SFT, it consistently and largely improves the model performance within the 250

training steps. With the same training steps, DPO only yields marginal improvements, while SFT even demonstrates a decline in performance. We provide the total training duration in Appendix D. Such a great improvement can be largely attributed to the *EM* process shown in Figure 10. Notably, our approach exhibits a convergence boundary after approximately 250 steps.

## 7 CONCLUSION

Prior studies suggest that long-context models typically follow a *retrieval-then-generation* paradigm, where the "retrieval context" may be overwhelmed by excessive irrelevant tokens. To address this issue, we present a fine-grained analysis of contextual noise in long-context inputs. We introduce a novel metric, the IG score, to effectively identify critical tokens, and observe that reducing contextual noise enables models to focus more precisely on critical tokens. Building on these insights, we propose Context Denoising Training (CDT), a training strategy designed to both enhance the model's attention to critical tokens and strengthen the association between salient tokens and the model prediction. Experiments across 4 task types (including both short and long context length) and different models demonstrate the superiority of our method. With CDT, an open-source 8B model can even achieve comparable performance with GPT-4o on real-world long-context tasks.

## ETHICS STATEMENT

We confirm that this work adheres to ethical research practices. All data and LLMs used are publicly available (including API format) and properly cited. No human subjects were involved. The Use of LLM statement is illustrated in Appendix H.

## ACKNOWLEDGEMENT

We want to thank all the anonymous reviewers for their valuable comments. This work was supported by the National Science Foundation of China (NSFC No. 62576232) and the Young Elite Scientists Sponsorship Program by CAST (2023QNRC001).

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

## A    ILLUSTRATION OF TRAINING EFFICIENCY OF CURRENT METHODS

Training efficiency comparisons across current long-context methods are inherently challenging: performance gains typically exhibit diminishing returns with increased token budgets under long-context training settings, and reported results often stem from divergent training setups — including data composition, optimization hyperparameters, and hardware configurations. These factors render direct "gain-per-token" comparisons unreliable when conditions are unmatched. To fairly compare training efficiency across methods — despite differing hyper-parameters and convergence behaviors — we adopt a controlled proxy: average task gain per 1B tokens, measured under identical data, optimizer, batch size, learning rate, and hardware ($8 \times$ A100 GPUs). Specifically, we compare ProLong (Gao et al., 2024b) - one long-context SFT method, and LongCE (Fang et al., 2024b) - one token-level re-weighting training method, on the Llama3-8B-Base model. As shown in Table 3, we evaluate model performance on LongBench-E (12 real-world tasks) per 50 training steps (0.41B tokens per 50 steps), and find that LongCE achieves a 3.7-point gain per 1B tokens versus ProLong's 1.8-point gain per 1B tokens.

Table 3: Performance comparison between ProLong (SFT) and LongCE across training steps, where each step contains the same training setting.

| Method | Step 0 | Step 50 | Step 100 | Step 150 | Step 200 |
|---|---|---|---|---|---|
| ProLong (SFT) | 25.50 | 27.32 | 28.15 | 28.44 | 29.13 |
| LongCE (same data) | 25.50 | 28.30 | 29.72 | 31.01 | 32.91 |

## B    PRELIMINARY STUDY DETAILS

### B.1    PRELIMINARY TASK CONSTRUCTION

**Task Selection**    We select 3-hop and 4-hop tasks based on qa3 tasks in the BABILong Benchmark to build our datasets, as these tasks generally pose significant challenges for LLMs. However, it is worth noting that the original BABILong qa4 samples do not truly require 4-hop reasoning to produce correct outputs. For example, a sample from this subset with 0k context is shown in Figure 11. In this case, the task only requires attention to a single fact, "The bedroom is west of the bathroom" to answer the question, while the first sentence serves as an interference fact. Even in terms of keywords, the model only needs to focus on three keywords: "bathroom", "west", and "bedroom" from the second sentence. Thus, we design our 4-hop dataset based on the BABILong qa3 source data, with one sample shown in Figure 12. By carefully arranging the order of facts and reducing the conditions of questions in the long context, we ensure that the model is required to search for all four supporting facts in sequence to produce the correct output.

Table 4: Variable settings, where R. denotes random.

| Hops | Samples | Permute | Lengths |
|---|---|---|---|
| 2 | 100 | 5 | 8K |
| 3/4 | R. | R. | 0k - 64k |

**Controlled Evaluation Data Synthesis**    We use the 4-hop task with non-zero context as an example here. As shown in Table 4, all variables used for building data include the facts sample, the facts permutation, and the context length. Firstly, we select source samples from the BABILong official file "qa3_three-supporting-facts" as our base data. Then, we modify the original BABILong qa3 supporting facts following the pattern shown in Figure 13. Afterward, we add interference to these four original facts while maintaining the relative order of the supporting facts. The process begins by selecting a noise context of the appropriate length and inserting the facts into it. Specifically, we divide the noise context into 10 equal-length chunks, leaving 10 candidate positions for the insertion of the 4 supporting facts (excluding the tail). Next, we randomly select five permutations from the

Table 5: Performance statistics of using different numbers of attention heads on our preliminary synthetic task. Notably, we find that selecting the top-30 heads yields results that are nearly identical to those obtained when using all attention heads.

| Head Number | Supporting | | Interference | | Irrelevant | | Low-frequency | |
|---|---|---|---|---|---|---|---|---|
| | Correct | Wrong | Correct | Wrong | Correct | Wrong | Correct | Wrong |
| Top-30 | 0.21 | 0.11 | 0.07 | 0.17 | 0.72 | 0.72 | 0.00 | 0.00 |
| All | 0.20 | 0.13 | 0.09 | 0.15 | 0.71 | 0.72 | 0.00 | 0.00 |

full set of $C_{10}^4$ candidate position permutations. After injecting noise, we randomly insert interference facts, i.e., facts that are similar to the supporting facts but irrelevant, among all sentences. We ensure that at least one interference fact is placed after the last supporting fact to test the model's robustness. To ensure the correctness of the samples, we make sure that the objects appearing in the interference facts do not overlap with those in the supporting facts. Additionally, we ensure that the number of interference facts is between one and two times the number of supporting facts to avoid making the samples either too easy or too difficult. Finally, for all samples with the same context length, we use the same noise context to maintain consistency. In the end, we randomly insert a few emojis into the constructed context to test the sensitivity of the model to low-frequency tokens. For the 3-hop task, we directly use the original qa3 task format from BABILong as the base, and the subsequent processing follows a similar approach to the one described above for the 4-hop task.

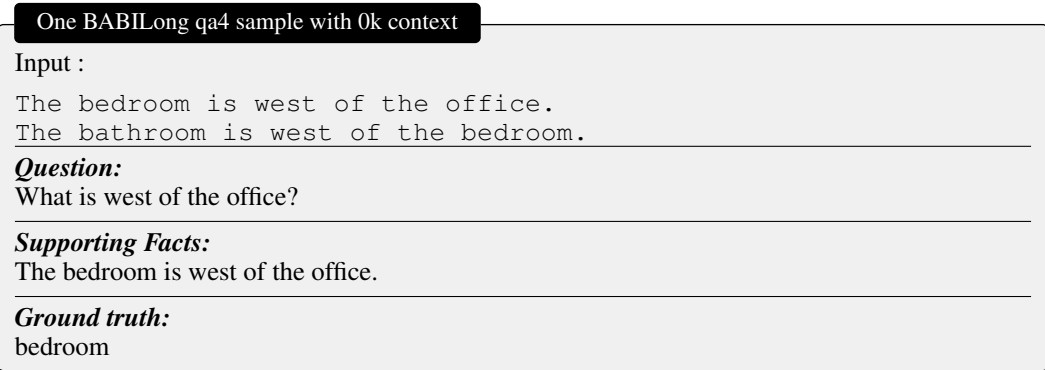

Figure 11: A BABILong qa4 sample with 0k context

## B.2 DESIGN OF IG SCORE

Prior work (Wu et al., 2024) has shown that not all attention heads behave uniformly, i.e., some are specialized for retrieval-like behaviors, while others are not. However, it is important to note that these findings are primarily derived from studies focused on copy-oriented tasks, such as NIAH. In contrast, our task involves reasoning and inference, which fundamentally differs from the objectives of retrieval heads. As a result, the mechanisms for attending to relevant context in our setting cannot be directly aligned with those used in retrieval-focused tasks. To further analyze the appropriate number of attention heads to select, we conduct experiments where we select the top-k attention heads (k = 30) that retrieve the most relevant information based on the attention scores (Table 5). We find that the performance using only a subset of attention heads was highly consistent with the results obtained by averaging over all attention heads. Therefore, for simplicity and ease of deployment, we adopt the latter approach, i.e., averaging IG scores across all attention heads.

```
One of our 4-hop samples with 0k context
```

Input :

```
Mary journeyed to the office.
Mike went to the office.
Mary got the apple.
Daniel picked up the football.
Daniel went back to the bedroom.
Mary journeyed to the bathroom.
Mary dropped the apple.
Jonh went to the bathroom.
```

*Question:*
Where was the apple's location prior to the place where the apple was discarded, left or dropped?

*Supporting Facts:*
Mary journeyed to the office.
Mary got the apple.
Mary journeyed to the bathroom.
Mary dropped the apple.

*Ground truth:*
office

Figure 12: One of our 4-hop samples with 0k context

```
The pattern of our 4-hop sample
```

```
Supporting fact1: {x} {m} the {y1}
Supporting fact2: {x} {p} the {o}
Supporting fact3: {x} {m} the {y2}
Supporting fact4: {x} {d} the {o}
```

*Question:*
Where was the **{o}**'s location prior to the place where the **{o}** was discarded, left or dropped?

*Ground truth:*
**{y1}**

*Explanation:*
**{x}** : a character name, selected from {Mary, Daniel, Mike, ...}
**{m}** : a predicate indicating movement, selected from {went to, journeyed to, travelled to, ...}
**{y1}, {y2}** : two different locations, selected from {office, bedroom, bathroom, ...}
**{p}** : a predicate indicating picking up, selected from {picked up, took, grabbed, ...}
**{d}** : a predicae indicating dropping, selected from {dropped, put down, discarded, ...}
**{o}** : an object name, selected from {apple, football, milk, ...}

Figure 13: The pattern of our 4-hop sample

## C  DERIVATION OF RELATION BETWEEN INFORMATION FLOW AND EMBEDDING GRADIENTS

In transformer-based models, the Information Flow in attention is essentially the product of the attention distribution and its corresponding gradient. Therefore, we can transform the derivation into **constructing the gradient relationship between the attention score distribution ($A$) and the embedding ($E(X)$)**. This can be established via the chain rule and implemented through the specific computation steps of the attention mechanism. Notably, in the following derivation, for simplicity, we omit the activation layers in the model. Additionally, considering that transformer-based models are composed of multiple identical network blocks stacked together, one can easily

extend the conclusions from a single layer to multiple layers. Therefore, we focus on proving the case with **one embedding layer and one attention module**.

Given the basic definition of the attention mechanism, we have:

$$
\begin{cases}
Q = E(X)W_Q, & A = \text{softmax}\left(\frac{QK^T}{\sqrt{d}}\right), \\
K = E(X)W_K, & O = A \cdot V, \\
V = E(X)W_V,
\end{cases}
$$

where $W_Q, W_K, W_V \in \mathbb{R}^{d \times d}$ are the model parameters, $O$ is the attention output, $E(X) \in \mathbb{R}^{n \times d}$ is the input embedding matrix, $n$ and $d$ are sequence length and model dimension, respectively.

Let the loss function be $L$. By the chain rule, the gradient of the loss with respect to $E(X)$ is:

$$
\frac{\partial L}{\partial E(X)} = \frac{\partial L}{\partial O}\frac{\partial O}{\partial E(X)} = \frac{\partial L}{\partial A}\frac{\partial A}{\partial E(X)} \\
+ \frac{\partial L}{\partial V}\frac{\partial V}{\partial E(X)}. \tag{5}
$$

Since we have $\frac{\partial V}{\partial E(X)} = W_V^T$ and $\frac{\partial O}{\partial V} = A$, the gradient relationship between $A$ and $E(X)$ is:

$$
\frac{\partial L}{\partial E(X)} \propto \frac{\partial L}{\partial A}\frac{\partial A}{\partial E(X)} \tag{6}
$$

To eliminate the influence of the $\text{Softmax}(\cdot)$ function, we can further decompose equation 6 into:

$$
\begin{cases}
S = \dfrac{QK^T}{\sqrt{d}}, \\
\dfrac{\partial L}{\partial E(X)} \approx \dfrac{\partial L}{\partial A} \cdot \left(\dfrac{\partial A}{\partial S} \cdot \dfrac{\partial S}{\partial E(X)}\right),
\end{cases} \tag{7}
$$

where $\frac{\partial A}{\partial S}$ is the Jacobian of $\text{Softmax}(\cdot)$ function, with elements $A_{ij}\left(\delta_{ik} - A_{ik}\right)$[3].

For each element $S_{ij} = \frac{Q_i K_j^T}{\sqrt{d}} \in S$, the gradient with respect to $E(X)$ can be written as:

$$
\frac{\partial S_{ij}}{\partial E(X)} = \frac{\partial \left(\frac{(E(X)_i W_Q)(E(X)_j W_K)^T}{\sqrt{d}}\right)}{\partial E(X)} \\
= \frac{1}{\sqrt{d}}\left(W_Q^T \cdot K_j \cdot \delta_{ik} + W_K^T \cdot Q_i \cdot \delta_{jk}\right). \tag{8}
$$

Based on equation 7 and equation 8, we can summary that:

$$
\frac{\partial L}{\partial E(X)_i} \propto \underbrace{\frac{\partial L}{\partial A_{ij}}}_{\text{Sensitivity of } L \text{ to } A} \\
\times \underbrace{A_{ij}(1 - A_{ij})}_{\text{Derivation from Softmax}} \\
\times \underbrace{\frac{\partial S_{ij}}{\partial E(X)}}_{\text{Linear Transformation}}. \tag{9}
$$

Based on equation 9, we can derive that when $A_{ij}$ increases, indicating higher attention between token $i$ and token $j$, the sensitivity of $L$ to $A$ ($\frac{\partial L}{\partial A_{ij}}$) also increases. This results in larger derivatives

---

[3]$\delta_{ik}$ is the Kronecker delta function. If $i$ equals to $k$, $\delta_{ik} = 1$, else $\delta_{ik} = 0$. We can also rewrite this equation into $A_{ij}(1 - A_{ij})$.

Table 6: Configuration of context window scaling training setting.

| Context Window Scaling Training Setting | |
| --- | --- |
| Backbone | Llama-3-8B-base |
| Training Objective | Language modeling |
| RoPE base | 20,000,000 |
| Context window size | 8K → 64K |
| Data seq-length | 64,000 |
| Deepspeed | Zero2 |
| Global batch size | 64 |
| Epoch | 2 |
| Training Steps | 160 |
| Ring-attention size | 4 |
| Learning-rate | 1e-5 |
| LR-scheduler | cosine_with_min_lr |
| Optimizer | Adam ($\beta_1 = 0.9, \beta_2 = 0.95$) |
| GPUs | A100 (80GB) $\times$ 8 |
| Training time | $\approx$8h / epoch |
| Training data | PG19 (Rae et al., 2019) |
| Total consumed tokens | 0.65B |

Table 7: Configuration of language modeling training setting.

| Language Modeling Post-training Setting | |
| --- | --- |
| Backbone | Llama-3.1-8B-base |
| Training Objective | Language modeling |
| RoPE base | 500,000 |
| Context window size | 128K |
| Data seq-length | 64,000 |
| Deepspeed | Zero2 |
| Epoch | 2 |
| Global batch size | 32 |
| Training Steps | 320 |
| Ring-attention size | 4 |
| Learning-rate | 1e-5 |
| LR-scheduler | cosine_with_min_lr |
| Optimizer | Adam ($\beta_1 = 0.9, \beta_2 = 0.95$) |
| GPUs | A100 (80GB) $\times$ 8 |
| Training time | $\approx$8.5h / epoch |
| Training data | PG19 (Rae et al., 2019) |
| Total consumed tokens | 0.65B |

Table 8: Configuration of long-context SFT training setting.

| Long-context Alignment Training Setting | |
| --- | --- |
| Backbone | Llama-3.1-8B-Instruct |
| Training Objective | Supervised fine-tuning |
| RoPE base | 500,000 |
| Context window size | 128K |
| Data seq-length | 4,000~128,000 |
| Deepspeed | Zero2 |
| Global batch size | 32 |
| Epoch | 2 |
| Training Steps | 250 |
| Ring-attention size | 4 |
| Learning-rate | 1e-5 |
| LR-scheduler | cosine_with_min_lr |
| Optimizer | Adam ($\beta_1 = 0.9, \beta_2 = 0.95$) |
| GPUs | A100 (80GB) $\times$ 8 |
| Training time | $\approx$6.5h / epoch |
| Training data | LongMIT (Chen et al., 2024b), LongAlpaca (Chen et al., 2023c) |
| Total consumed tokens | 0.53B |

Table 9: Testing configuration of RULER

| Evaluation Configuration of RULER | |
| --- | --- |
| Question Answering | qa_1, qa_2 |
| Single NIAH | niah_single_1, niah_single_2, niah_single_3 |
| Multi-keys NIAH | niah_multikey_1, niah_multikey_2, niah_multikey_3 |
| Multi-values NIAH | niah_multiquery |
| Multi-queries NIAH | niah_multivalue |
| Others | common words extraction (CWE), frequent words extraction (FWE), variable tracking (VT) |
| Length | 32K, 64K |
| Num samples/task | 50 |

on the embeddings. Additionally, if $A_{ij}$ becomes excessively large, approaching 1, the value of $A_{ij}(1 - A_{ij})$ might tend toward 0. However, this is often not an issue in long-context scenarios, as the attention scores are unlikely to approach values near 0.5 due to the long context. Even if they exceed 0.5 (possibly for some special tokens), the increase in the first term ($\frac{\partial L}{\partial A_{ij}}$) helps mitigate this effect.

# D  IMPLEMENTATION DETAILS

## D.1  TRAINING DETAILS

For all experiments, we utilize the open-source training framework OpenRLHF[4] (Hu et al., 2024), Ring-flash-attention[5] (Liu et al., 2023) and DeepSpeed (Rajbhandari et al., 2020). For LongCE training (Fang et al., 2024b), we set the sliding context window size as 8192 and employ the recommended hyper-parameters in the official code [6].

---

[4] https://github.com/OpenRLHF/OpenRLHF.git
[5] https://github.com/zhuzilin/ring-flash-attention.git
[6] https://github.com/PKU-ML/LongPPL.git

Table 10: Testing configuration of BABILong.

| Metric | QA1 | QA2 | QA3 | QA7 | QA8 |
|---|---|---|---|---|---|
| Num | 100 | 100 | 100 | 100 | 100 |
| Supporting Fact | 1 | 2 | 3 | 1∼10 | 1∼8 |
| Interference Fact | 1∼9 | 1∼66 | 1∼317 | 1∼42 | 1∼42 |

Table 11: Evaluation results on HELMET (Yen et al., 2025).

| Model | Recall | RAG | ICL | Re-rank | QA | Summ. | Cite | Avg. |
|---|---|---|---|---|---|---|---|---|
| Claude-3.5-Sonnet | 94.7 | 38.1 | 61.0 | 7.2 | 12.6 | 36.6 | 18.7 | 38.4 |
| Mistral-Nemo-12B | 14.6 | 40.0 | 84.0 | 0.0 | 22.5 | 18.5 | 0.5 | 25.7 |
| ProLong-512K-Instruct | 98.8 | 63.2 | 86.5 | 22.5 | 43.9 | 29.2 | 1.4 | 49.4 |
| Meta-Llama-3.1-8B | 95.2 | 59.5 | 83.9 | 14.0 | 43.2 | 27.0 | 2.9 | 46.5 |
| + CDT | 97.2 | 61.8 | 86.6 | 18.5 | 46.7 | 27.9 | 9.4 | **49.7** |

**Context Window Scaling**     To scale the context window size of the Llama-3-8B-base model from 8K to 64K (8×), we adjust the RoPE base from 500,000 to 20,000,000 and directly train the model. We provide training configurations in Table 6.

**Data Post-processing Details**     For the context window scaling experiments, we employ the PG-19 (Rae et al., 2019) dataset. For long-context SFT and CDT experiments, we construct our data from publicly available long-context QA datasets, including LongMiT (Chen et al., 2024b) and LongAlpaca (Chen et al., 2023c). The LongMiT dataset primarily consists of multi-hop QA tasks that require reasoning over 2 to 6 evidence passages. To adapt it for our setting, we apply two pre-processing steps: (i) Length distribution control — we constrain the sampled instances to fall within 16K–128K tokens. This range balances the need for sufficiently long contexts with training efficiency, given our compute resources (8 × A800 GPUs). Excessively long sequences were avoided as they considerably slow down training. (ii) Evidence balancing — we uniformly sample across different numbers of supporting passages to obtain a more balanced distribution for multi-hop reasoning. To complement this, we include data from LongAlpaca, which predominantly features single-evidence QA with lengths around 16K tokens (under our model's tokenizer). This addition enriches the training distribution by covering shorter single-evidence scenarios, which are underrepresented in LongMiT. In total, our final training set comprises 7,000 samples from LongMiT and 1,000 samples from LongAlpaca, which are shuffled together before training.

**Language Modeling Post-training and Long-context SFT**     The language modeling post-training and long-context SFT are directly applied to the Llama3.1-8B-base and Llama3.1-8B-Instruct, respectively, which already have 128K context window size. We provide the training configurations in Table 7 and Table 8 respectively.

## D.2    Evaluation Details

We conduct long-context evaluation mainly based on the long-context evaluation framework `LOOM-Eval`[7] (Tang et al., 2025).

**HELMET**     HELMET (Yen et al., 2025) is a comprehensive long-context evaluation benchmark containing 7 different subtasks, including recall, RAG, in-context learning (ICL), re-rank, QA, summarization, and citation. The context length of test samples ranges from 0 to 128K tokens. For inputs exceeding 128K, we truncate from the end to fit within the model's maximum context window. We show the experimental results on HELMET in Table 11, where our CDT model achieves the best performance.

---

[7]https://github.com/LCM-Lab/LOOM-Scope

Table 12: Evaluation results of two more LLMs on real-world long-context tasks and long-form reasoning tasks.

| Models | | LongBench-E | | | | | | BABILong |
|---|---|---|---|---|---|---|---|---|
| | Type | S-Doc QA | M-Doc QA | Summ | Few-shot | Code | Avg. | Avg. |
| Qwen2.5-7B-Instruct | - | 44.54 | 46.29 | 28.15 | 56.03 | 16.52 | 38.30 | 43.32 |
| + CDT | SFT | **44.93** | **47.29** | **28.65** | **57.33** | **19.18** | **39.48** | **47.56** |
| Qwen3-8B | - | 44.12 | 48.10 | 29.30 | 44.12 | 29.18 | 38.85 | 48.06 |
| + CDT | SFT | **45.33** | **49.13** | **31.89** | **46.24** | **32.98** | **41.11** | **52.88** |
| Mistral-V0.3-Instruct | - | 44.89 | 40.76 | 20.52 | 67.11 | 47.04 | 44.06 | 22.36 |
| + CDT | SFT | **45.01** | **41.79** | **26.08** | **67.75** | **57.27** | **47.58** | **53.84** |

**LongBench-E** LongBench-E is a variant of LongBench (Bai et al., 2024b) designed specifically for long-context real-world tasks. We chose LongBench-E because it shares the same test dataset distribution as LongBench while covering a wider range of context lengths. For the Llama3-8B-base model, we truncate the input to 8K tokens, whereas for other models, we truncate the input to 32K tokens.

**Language Modeling** For the language modeling task, we calculate both LongPPL and PPL metrics on the GovReport dataset (Huang et al., 2021), which consists of long sequences from government reports. We sample 50 documents from GovReport, each with a context length of up to 32K tokens.

**RULER** RULER (Hsieh et al., 2024) is a comprehensive synthetic dataset that includes 6 different testing categories to evaluate a model's long-context understanding capabilities. We utilize all test categories, with each category containing 50 test samples covering lengths of 32K and 64K. We post the testing configuration of RULER in Table 9.

**Long-form Reasoning** We evaluate the long-form reasoning capability of models on selected tasks from BABILong (Kuratov et al., 2024). Specifically, we select tasks that involve multiple supporting facts, as well as QA1, as the testing dataset. The BABILong testing configurations are shown in Table 10.

### D.3 BASELINE ILLUSTRATION

We evaluate our method on three foundation models, i.e., LLaMA-3-8B-Base, LLaMA-3.1-8B-Base, and LLaMA-3.1-8B-Instruct—to ensure fair and consistent comparisons across all baselines. The baselines include: YaRN, which extends the context window using an improved NTK-based positional scaling method; CE (Cross Entropy), a standard language modeling objective without any context-aware weighting; LongCE, which builds upon the LongPPL method by identifying key tokens via perplexity during training and assigning them higher loss weights; SFT, an instruction tuning setup where input tokens are excluded from the loss calculation; and LOGO, a DPO-based training approach designed to mitigate misalignment in long-context tasks. Additionally, we compare against several strong open-source long-context models: ProLong-512K-Instruct and NExtLong-512K-Instruct, which apply long-context scaling techniques on top of LLaMA-3-8B-Instruct and LLaMA-3.1-8B-Instruct, re-

Table 13: Model performance on language modeling tasks.

| Models | LongPPL | PPL |
|---|---|---|
| Llama-3-8B-Base | > 100 | > 100 |
| + YaRN | 3.55 | 5.60 |
| + CE | 3.90 | 6.46 |
| + LongCE | 3.55 | 5.60 |
| + CDT (ours) | **3.04** | **5.40** |
| Llama-3.1-8B-Base | 3.22 | **4.79** |
| + CE | 3.28 | 4.86 |
| + LongCE | 3.24 | 5.28 |
| + CDT (ours) | **2.10** | 5.19 |
| Llama-3.1-8B-Instruct | 4.05 | **5.52** |
| + SFT | 3.31 | 5.51 |
| + LOGO | 4.11 | 5.54 |
| + CDT (ours) | **2.36** | 5.64 |

Table 14: Statistical significance calculation on LongBench-E data with t-Test.

| Models | P-Value |
|---|---|
| Llama3-8B-Base V.S. Llama3-8B-Base-CDT | 3.68e-15 |
| Llama3.1-8B-Base V.S. Llama3.1-8B-Base-CDT | 1.53e-2 |
| Llama3.1-8B-Instruct V.S. Llama3.1-8B-Instruct-CDT | 2.39e-3 |

spectively; and LLaMA-3.1-8B-SEALONG, a DPO-trained model specifically optimized for long-context alignment.

## E  MORE EVALUATION RESULTS

### E.1  ANALYSIS OF RESULTS ON REAL-WORLD LONG-CONTEXT TASKS

The strong performance of CDT on code-related tasks, as shown in Table 1, is particularly notable. Code Completion requires models to accurately interpret local context and predict missing segments accordingly. CDT is especially well-suited for such tasks, as it enhances the model's ability to focus on local context information during generation, which likely contributes to the observed performance improvements. Table 15 offers a more intuitive illustration through a specific Code Completion example. In LongBench-E, this task is evaluated using the Edit Similarity (Edit Sim) metric, which is highly sensitive to the number of tokens generated—especially under the official 64-token generation limit. In the provided example, LLaMA-3.1-8B-Instruct produces entirely incorrect outputs, while GPT-4o generates overly lengthy responses that negatively affect the Edit Sim score. In contrast, the CDT-enhanced model generates a concise and accurate response, resulting in a significantly higher Edit Sim score. Furthermore, CDT leads to substantial improvements for both LLaMA-3.1-8B-Instruct and LLaMA-3-8B-Base on the Code task. These improvements can be attributed to two main factors. First, the training set includes code completion instances (e.g., 263 examples from LongMIT), which enable the model to learn relevant instruction-following patterns. Second, the baseline model's lower performance in this domain makes the gains from CDT more apparent. By contrast, LLaMA-3.1-8B-Base already demonstrates strong performance on code-related tasks—likely due to the composition of its pretraining data—resulting in smaller relative gains when CDT is applied.

### E.2  GENERALIZING CDT TO MORE MODELS

We apply our CDT method to more LLMs, including Qwen2.5-7B-Instruct (Yang et al., 2024) and Mistral-V0.3-Instruct (Jiang et al., 2023). We evaluation the model performance on real-world long-context tasks, long synthetic tasks, and long-form reasoning tasks. We report the model performance in Table 12, where we can observe that our CDT can significantly improve the model performance on different models. For instance, the Mistral-V0.3-Instruct model obtains more than 30 points on the long-form reasoning task.

### E.3  EVALUATION RESULTS ON LANGUAGE MODELING TASKS

Apart from evaluating with LongPPL on the language modeling task, we also calculate the PPL scores, which are shown in Table 13.

### E.4  EXPERIMENT STATISTICAL SIGNIFICANCE

We collect the prediction results of the original model and the CDT model on the LongBench-E benchmark, and conduct a paired-samples t-test to assess the statistical significance of the mean difference before and after the improvement, shown in Table 14. The results show that our method significantly outperforms the baseline model at the 5% significance level, indicating that our method achieves statistically significant improvements.

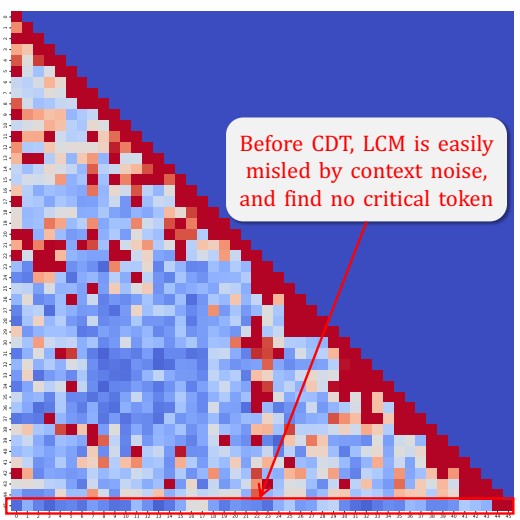
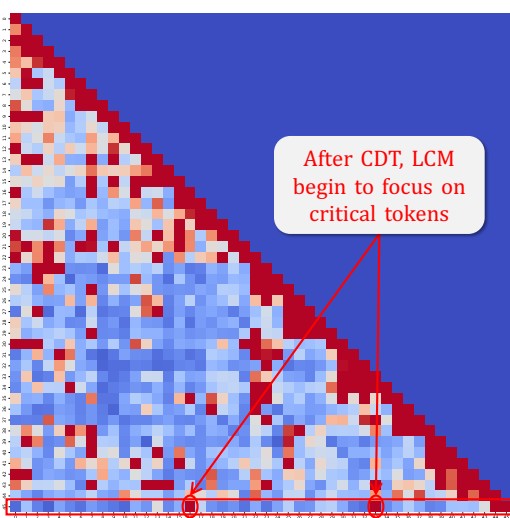

(a) Attention map of 24th layer before CDT.

(b) Attention map of 24th layer after CDT.

Figure 14: Comparison between Attention Maps Before and After CDT. In each figure, a deeper (red) color indicates larger model attention to the corresponding context chunk. The final row of each map represents how the question attends to the entire input sequence, including both the context and the question itself. For clearer visualization, we recommend zoom in on this figure.

## F ANALYSIS OF ATTENTION MAP BEFORE AND AFTER CDT

In this section, we present a visualization of the model's attention patterns before and after applying the CDT training strategy. Given the long input length (12,000 tokens) used in our evaluation, we evenly partition the input sequence into 46 chunks and calculate the total attention score for each chunk individually. For each chunk, a higher total attention score indicates that the model places greater focus on this chunk. We visualize the attention maps of the 24th layer of the model, as this layer provides the clearest representation of CDT's impact. As shown in Figure 14, we can observe that, before applying CDT, the model's attention is predominantly concentrated on the question itself (the rightmost portion of the final row in Figure 14a), while key information within the context is largely overwhelmed by noise. In contrast, after CDT training, the model not only attends to the question but also shows significantly increased attention to relevant contextual information, as highlighted by the red circles in the final row of Figure 14b. **It is noteworthy that the attention map shows no significant changes before and after CDT training, indicating that CDT training does not compromise the original characteristics of the LCM. Instead, it enhances the ability of LCM to capture critical information.**

## G LIMITATION AND FUTURE WORK

Due to the expectation maximization (EM) nature of CDT, it includes an additional context noise detection process, which introduces extra computational costs during the training phase. Although we have demonstrated in Section 6.3 that these additional costs are negligible compared to the performance gains, theoretically, *the noise detection cost will increase as the model size grows since it involves a complete forward-backward propagation process*. We leave this for future work, aiming to explore a simpler method for identifying the context noise or to develop more efficient model architectures. For example, designing specific network modules to handle noise, as proposed in Ye et al. (2024), could be a promising direction. Additionally, we observe that *the improvement brought by our method on complex reasoning tasks is not as significant as that on other tasks*, and we are yet to understand the relationship between this and the training data or the training objective function. In the future, we aim to further investigate the impact of context noise on the model's long-form reasoning abilities, as well as the relationship between the CDT strategy and the enhancement of the model's reasoning capabilities.

## H  USE OF LLMS

During the writing of this paper, we leveraged large language models (LLMs) to refine the clarity and fluency of our writing, particularly in the Abstract and Introduction sections. Specifically, we used the Qwen web interface [8] to access the Qwen series of models (e.g., Qwen-Max), inputting early drafts of these sections and requesting stylistic improvements while preserving technical accuracy and original intent. The model's suggestions helped enhance sentence structure, academic tone, and overall readability. All final content was carefully reviewed, validated, and edited by the authors to ensure fidelity to our research and adherence to scholarly standards.

## I  ERROR ANALYSIS

In this section, we analyze the error pattern of partial model predictions on real-world long-context tasks. As shown in Table 15, we use colored text to highlight the correct and incorrect parts of the model's predictions.

---

[8] https://chat.qwen.ai

Table 15: Generation samples of Llama3.1-8B-Instruct, Llama3.1-8B-Instruct-CD (ours), and GPT4o on real-world long-context tasks.

| Code Completion Task |
|---|

**Question**: Completing the code based on the context: [ ··· ]\n self.interface = self.interface.replace(' ', ").lower() \n self.param_type = None \n self.if_type = None \n # state \n self.results = dict() \n [ Completion ] self.result = dict() \n def check_params(self): \n """Check all input params""" \n if not self.interface: \n self.module.fail_json(msg='Error: Interface name cannot be empty.')[ ··· ]

**Llama3.1-8B-Instruct**: for (int i = 0; i < s.length(); i++) {if (!Character.isDigit(s.charAt(i)) && s.charAt(i) != ',') }
**Llama3.1-8B-CD (ours)**: if self.interface == 'all':\n self.param_type = INTERFACE_ALL\n
**GPT4o**: if self.interface.lower() == 'all':\n self.param_type = INTERFACE_ALL\n elif '/' in self.interface or self.interface.isdigit():\n self.param_type = INTERFACE_FULL_NAME\n else:\n self.if_type = get_interface_type(self.interface)\n
**Golden Answer**: if self.interface == 'all':\n

| Long-context Multi-hop Reasoning Task |
|---|

**Question**: What did M. King Hubbert's employer announce it was in the process of doing in April 2010? [ ··· ] In April 2010 Shell announced that the corporation is in process of trying to find a potential buyer for all of its operations in Finland and is doing similar market research concerning Swedish operations. [ ··· ]

**Llama3.1-8B-Instruct**: Shell announced the sale of some of its assets, including its liquefied petroleum gas (LPG) business, to meet the cost of a planned $28
**Llama3.1-8B-CD (ours)**: trying to find a potential buyer for all of its operations in Finland and is doing similar market research concerning Swedish operations.
**GPT4o**: MV Miralda's owner announced it was in the process of trying to find a potential buyer for all of its operations in Finland and in doing similar market research concerning.
**Golden Answer**: trying to find a potential buyer for all of its operations in Finland

| Single-QA Long-context Task |
|---|

**Question**: What metrics are used in evaluation? [ ··· ] Figure FIGREF22 shows the results of these experiments wherein we are able to maintain the **precision** on the subset of the training set used for testing while reducing its **recall** and thus generalising the model. The fact that the model is generalizing on a dissimilar dataset is confirmed by the increase in the development set. **F1 score**. [ ··· ]

**Llama3.1-8B-Instruct**: Macro-averaged F-score.
**Llama3.1-8B-CD (ours)**: macro-averaged F-score, precision, recall, F1 score.
**GPT4o**: Macro-averaged F-score.
**Golden Answer**: precision, recall , F1 score

