# OpenReview forum: "Revisiting Long-context Modeling from Context Denoising Perspective"
_ICLR.cc/2026/Conference — ICLR 2026 Poster_

### Official Review · Reviewer_Rdcm · 2025-10-18

**Soundness:** 4
**Presentation:** 3
**Contribution:** 3
**Rating:** 8
**Confidence:** 3

**Summary:**

This paper introduces Context Denoising Training (CDT), a straightforward yet effective training strategy that improves attention on critical tokens while reinforcing their influence on model predictions. The authors first propose the Fact Retrieval (FR) score and Integrated Gradient (IG) score to identify and quantify context noise, demonstrating that existing long-context methods fail to distinguish critical from irrelevant information, degrading performance. They show that simple mitigation of detected context noise can substantially boost the model’s attention on critical tokens and benefit subsequent predictions.

Building on this insight, CDT subtracts the corresponding gradients to manipulate the irrelevant token embeddings, thus suppressing the context noise. This allows the model to focus on critical tokens and strengthens the causal link between them and the final output. Across LongBench, RULER, BABILong, LongPPL, and other benchmarks, CDT consistently boosts long-context performance.

**Strengths:**

1. This paper conducts extensive experiments across multiple settings and long-context benchmarks, thoroughly demonstrating the effectiveness of CDT over multiple baselines.
2. The introduction of the FR score, IG score, and identifier $\mathcal{I}(x_i)$ provides valuable insights into how long-context LLMs comprehend long contexts.
3. Beyond presenting an effective long-context training paradigm, the authors offer in-depth discussions and visualized analyses that solidify the credibility of their approach.

**Weaknesses:**

See Questions.

**Questions:**

1. The paper introduces the FR score, IG score, and an identifier. However, only the identifier is actually used in CDT. The FR and IG scores serve only as diagnostic tools because they require prior knowledge of the four token types in the context. I would appreciate it if the authors could clarify these two distinct contributions more clearly.
2. It is unclear why low-frequency words should be singled out. No prior work is cited to justify this taxonomy. A simpler partition into supporting, interference, and irrelevant tokens appears sufficient, and the proposed identifier itself only decides whether a token is irrelevant.
3. I find it confusing that authors compare CE, LongCE, and CDT with YaRN. YaRN is not a training paradigm. It is merely an interpolation method like NTK. Likewise, comparing CDT at inference time with FlexPrefill and XAttention is also confusing. These techniques aim at enhancing efficiency, not performance.
4. As a work focused on long-context training, CDT lacks citations to early efforts in this line[1-3]. In addition, the strategy of identifying critical tokens to improve training or inference should be clearly differentiated from similar techniques originally developed in short-input or long-output scenarios[4-5].
5. The authors note in the appendix that the improvement brought by our method on complex reasoning tasks is not as significant as that on other tasks. Since CDT also identifies critical tokens to enhance reasoning, could the high-entropy token approach[5] be combined to further boost performance?
6. Typo: irrelevant instead of irrevelant in Figure 3a

[1] Effective Long-Context Scaling of Foundation Models https://arxiv.org/abs/2309.16039

[2] Long Context is Not Long at All: A Prospector of Long-Dependency Data for Large Language Models https://arxiv.org/abs/2405.17915

[3] LongWanjuan: Towards Systematic Measurement for Long Text Quality https://arxiv.org/abs/2402.13583

[4] RHO-1: Not All Tokens Are What You Need https://arxiv.org/abs/2404.07965

[5] Beyond the 80/20 Rule: High-Entropy Minority Tokens Drive Effective Reinforcement Learning for LLM Reasoning https://arxiv.org/abs/2506.01939

---

> ### Author Response · Authors · 2025-11-18
> **Response to Reviewer Rdcm (Part I)**
>
> We sincerely appreciate your positive assessment of our work and are glad that the distinctions among the FR score, IG score, and the identifier are clearly understood by you! Below, we carefully respond to your concerns / questions one by one.
>
> ---
>
> **Question 1**: ``The paper introduces the FR score, IG score, and an identifier. However, only the identifier is actually used in CDT. The FR and IG scores serve only as diagnostic tools because they require prior knowledge of the four token types in the context. I would appreciate it if the authors could clarify these two distinct contributions more clearly.``
>
> **Response 1**:  Thank you for pointing out the need to clarify these two different contributions. This clarification is very important! To make this distinction clearer to readers, **we have added an explicit explanation in the Introduction (Lines 74–77)**, where we clarify that CDT uses gradients on token embeddings as the identifier during training, rather than directly relying on the IG score.
>
> ---
>
> **Question 2**: ``It is unclear why low-frequency words should be singled out. No prior work is cited to justify this taxonomy. A simpler partition into supporting, interference, and irrelevant tokens appears sufficient, and the proposed identifier itself only decides whether a token is irrelevant.``
>
> **Response 2**: Thank you for highlighting this ambiguity. We realize that the motivation for introducing “low-frequency words” was not sufficiently clear in the original manuscript. In Figure 2, “low-frequency words” refer specifically to tokens that appear extremely infrequently in the pre-training corpus and therefore exhibit very sparse distributional statistics. These are not simply uncommon words, but tokens that the **model has very limited exposure to during training, making them unlikely to provide meaningful semantic support for the prediction**. Our intent in isolating this category was to investigate whether the model displays abnormally high attention toward such very rare tokens when they appear in a long context. By inserting these tokens into synthetic contexts, we can test whether the model disproportionately attends to them despite their semantic irrelevance—thus offering additional evidence of the model’s susceptibility to contextual noise. **We also believe that some readers may naturally be curious about this aspect**, so we included it in the paper as an additional point of reference. Importantly, this choice does not affect the overall understanding of our method or conclusions. We hope this clarification helps, and we appreciate your understanding.
>
> ---
>
> **Question 3**: `I find it confusing that authors compare CE, LongCE, and CDT with YaRN. YaRN is not a training paradigm. It is merely an interpolation method like NTK. Likewise, comparing CDT at inference time with FlexPrefill and XAttention is also confusing. These techniques aim at enhancing efficiency, not performance.`
>
> **Response 3**: Thank you for raising this valuable point. We agree that YaRN, FlexPrefill, and XAttention are not training paradigms in the same sense as CE, LongCE, or CDT. Our intention was not to treat them as equivalent training methods, but rather to provide a comprehensive comparison across the broader landscape of long-context techniques, especially those that interact with or approximate the effects of denoising.
>
> The rationale for our baseline selection is that the denoising process in long-context modeling can be broadly abstracted as a form of token-wise reweighting within the loss or attention space.
>
> Under this view, we can have the following categories:
>
>   - **Binary (0–1) reweighting**.
>  Methods such as XAttention and key–value eviction mechanisms effectively remove certain tokens from the attention context, functioning as hard denoising.
>   - **Continuous/soft reweighting**.
>  Approaches like LongCE and our proposed CDT apply graded importance to tokens, representing soft denoising in the objective.
>   - **Standard objectives**.
> CE is included as a baseline that performs no reweighting, highlighting the benefit of explicit denoising mechanisms.
>
> Then, I will answer your questions one by one:
>
> **1. Why include YaRN?**
> Although YaRN is not a training paradigm, it is one of the most widely adopted testing-time scaling methods, and it has been reported to improve not only context length but sometimes task performance. Since YaRN can implicitly influence how models handle noise in extremely long contexts, we felt it was helpful to include it for completeness and for readers’ reference, while not positioning it as a direct comparison to CDT.
>
> > Due to character limitations, please refer to the response to Reviewer Rdcm (Part II) for the remaining reply regarding Weakness 3.”

---

> ### Author Response · Authors · 2025-11-18
> **Response to Reviewer Rdcm (Part II)**
>
> > Continual reply to weakness 3
>
> **2. Why mention FlexPrefill and XAttention?**
> While these methods mainly aim to improve efficiency at inference time, they also alter the effective context the model attends to (e.g., by discarding or compressing specific tokens). Since CDT addresses the effect of contextual noise, we believed it would be informative to compare against approaches that modify token influence during inference—even if their primary goal is efficiency rather than accuracy.
>
> **3. On additional baselines such as the Differential Transformer.**
>  We also explored including the Differential Transformer, which employs a related reweighting mechanism. However, publicly released implementations only support short contexts (≤8K for 8B models), and the original paper trains with a maximum context length of 2K over ~10B tokens. This setting is not directly comparable to our long-context training regimes (64K–128K, 300–500 steps), so such a comparison would be inherently unfair.
>
> Overall, our baseline selection aimed to **strike a balance between methodological relevance and practical feasibility, while giving readers a broad view of existing long-context techniques**. We genuinely welcome any further suggestions you may have—we are committed to ensuring that our evaluation remains fair, comprehensive, and informative.
>
> ---
>
> **Question 4**: `As a work focused on long-context training, CDT lacks citations to early efforts in this line[1-3]. In addition, the strategy of identifying critical tokens to improve training or inference should be clearly differentiated from similar techniques originally developed in short-input or long-output scenarios[4-5].`
>
> **Response 4**: Thank you very much for calling attention to these missing citations — we apologize for the oversight. After carefully reviewing the papers you suggested, we have incorporated them into the manuscript and clarified how they relate to our work.
> - **References [1–3]**, which focus on length-extrapolation and training strategies to extend context windows, have been added to Related Work Line 106-Line 125 (Length Extrapolation).
> - **References [4–5]**, which propose techniques for identifying or reweighting important tokens in short-input or long-output scenarios, have been added to Related Work Line 131-Line 133 (Token Importance and Denoising).
>
> We appreciate the pointer to these important works. Please let us know if there are any other references you believe are essential to include; we will gladly incorporate them.
>
> ---
>
> **Question 5**: `The authors note in the appendix that the improvement brought by our method on complex reasoning tasks is not as significant as that on other tasks. Since CDT also identifies critical tokens to enhance reasoning, could the high-entropy token approach[5] be combined to further boost performance?`
>
> [5] Beyond the 80/20 Rule: High-Entropy Minority Tokens Drive Effective Reinforcement Learning for LLM Reasoning https://arxiv.org/abs/2506.01939
>
> **Response 5**: Thank you for raising this thoughtful question. It touches on an important aspect regarding the **generalization of CDT to complex reasoning tasks**. As noted in the appendix, CDT brings less pronounced improvements on multi-step reasoning tasks compared to other benchmarks. We believe this stems from a fundamental characteristic of reasoning processes: **the token-to-prediction attribution is inherently less localized**.
>
> For instance, in a task such as BABILong, the reasoning chain may proceed as follows:
>
> > Bob is in the kitchen → Bob picks up an apple → Bob moves to the garden → Bob moves to the bedroom → Bob puts down the apple.
>
> When asked “Where is the apple?”, all intermediate steps contribute meaningfully to the final answer. However, CDT tends to **prioritize tokens whose gradients are most strongly aligned with the final prediction** — such as "Bob picked up the apple" and "Bob put down the apple" — while earlier steps in the chain may be incorrectly treated as noise.
> This explains why CDT provides weaker gains on tasks where all steps are semantically essential, but not equally attributable from the gradient perspective. Your suggestion to incorporate the high-entropy minority token strategy from Beyond the 80/20 Rule is indeed highly valuable. As shown in Section 2.1 (Eq. 1) of that paper, **token-level entropy provides an orthogonal signal that captures reasoning-relevant uncertainty**. Combining this entropy-based importance measure with CDT’s embedding-gradient signal could yield a more robust estimator of token criticality for long-chain reasoning. We believe such a hybrid approach could substantially improve CDT’s performance on complex reasoning tasks, and we are excited to explore this direction in future work.

---

> ### Author Response · Authors · 2025-11-18
> **Response to Reviewer Rdcm (Part III)**
>
> **Question 6**:  `Typo: irrelevant instead of irrevelant in Figure 3a`
>
> **Response 6**: Thank you for pointing out this typo. We have corrected “irrevelant” to “irrelevant” in Figure 3a (Line 162-171) in the revised version.
>
> ---
>
> **Summary of Rebuttal for Reviewer Rdcm**
>
> We sincerely appreciate your thoughtful feedback, which has helped us improve our manuscript. Below is a summary of the changes made in response to the reviewer’s concerns:
>
> - **Clarification of FR score, IG score, and identifier**:
>  We have clarified the distinct roles of the FR score, IG score, and the identifier in the paper. The identifier, used in CDT, leverages gradients on token embeddings during training, whereas the FR and IG scores serve as diagnostic tools, as explained in the Introduction (Lines 74–77).
>
> - **Justification for low-frequency words**:
>  We provided a more detailed explanation regarding the inclusion of low-frequency words in our analysis.
>
> - **Clarification on baseline comparisons**:
>  We clarified the rationale behind including YaRN, FlexPrefill, and XAttention in our baseline comparison. These techniques, though not training paradigms, are included to provide a comprehensive comparison of long-context techniques, especially those affecting token reweighting.
>
> - **Citations for early efforts in long-context training**:
>  We have added relevant citations to earlier works in the related literature, specifically regarding length extrapolation methods (Section 2.1) and techniques for identifying critical tokens (Section 2.2).
>
> - **Incorporating high-entropy token approach for reasoning tasks**:
>  We have discussed the potential for combining CDT with the high-entropy token approach to enhance performance on complex reasoning tasks. This direction will be explored in future work.
>
> - **Correction of typo in Figure 3a**:
>  We have corrected the typo from “irrevelant” to “irrelevant” in Figure 3a.
>
> ---
>
> We hope that these revisions address your concerns. We are grateful for your constructive suggestions, which have helped to strengthen the clarity and rigor of our work.
>
> If you have any further questions about our paper or our rebuttal responses, please do not hesitate to reach out; we would be very happy to clarify them for you!

---

> ### Comment · Reviewer_Rdcm · 2025-11-22
>
> Thank you for your detailed reply. One additional suggestion is to use gray text for the less critical supplementary comparisons (YaRN, FlexPrefill, and XAttention) to facilitate a comprehensive comparison and highlight the baseline that is mainly compared. Considering my score is already high enough, I've raised my confidence score as a recognition of your sincere response.

---

> > ### Author Response · Authors · 2025-11-23
> > **Thanks Reviewer Rdcm for supporting our paper!**
> >
> > Thank you very much for your support of our paper! We have used gray text for the less critical supplementary comparisons (YaRN, FlexPrefill, and XAttention) in Tables 1 and 2 to facilitate a comprehensive comparison and to better highlight the primary baseline under discussion. These changes have been implemented specifically at lines 343–344 and lines 388–389 in our revised manuscript!

---

### Official Review · Reviewer_L6Fs · 2025-10-30

**Soundness:** 3
**Presentation:** 3
**Contribution:** 3
**Rating:** 6
**Confidence:** 4

**Summary:**

This paper identifies a key weakness in long-context models (LCMs): their susceptibility to contextual noise—irrelevant tokens that distract the model from critical information. The authors propose a novel training strategy called Context Denoising Training (CDT), which improves a model’s ability to focus on salient tokens and strengthen their influence on predictions.

**Strengths:**

1. The Integrated Gradient (IG) score, which more accurately identifies important tokens compared to attention-based methods.

2. Manual noise suppression experiments: Showing that reducing noise in input embeddings boosts attention on critical tokens by ~10×.

3. The CDT training strategy: A lightweight, online method that detects and suppresses noisy tokens during training, improving model focus without heavy computational overhead.

4.Extensive evaluation: Across 4 task types (real-world, synthetic, language modeling, reasoning) and multiple models, CDT consistently outperforms baselines and even enables an 8B model to nearly match GPT-4o on LongBench-E.

**Weaknesses:**

1. Is there any relevant literature or experimental evidence supporting the statement in line 759 that "performance gains typically exhibit diminishing returns with increased token budgets"?

2. In lines 768–769, it is stated that "LongCE achieves a 13-point gain per 1B tokens versus ProLong’s 0.3-point gain per 1B tokens." How was the 13-point gain calculated?

3. Why are the results for NExtLong-512K-Instruct and ProLong-512K-Instruct shown in Table 1, but not included in Table 2?

**Questions:**

See Weaknesses.

---

> ### Author Response · Authors · 2025-11-18
> **Response to Reviewer L6Fs (Part I)**
>
> We are truly grateful for your careful and insightful review of our work! Below, we provide detailed responses to each of the points raised.
>
> ---
> **Weakness 1**: `Is there any relevant literature or experimental evidence supporting the statement in line 759 that "performance gains typically exhibit diminishing returns- with increased token budgets"?`
>
> **Response 1**: Thank you for pointing out this important issue! We agree that the original phrasing at Line 759 in the first version manuscript may have been misleading (becoming Line 813-814 in the latest manuscript version). Our intended meaning was that **under long-context training setting**, using increasingly more long-sequence data often leads to diminishing—and sometimes negative—returns. We have revised the text accordingly (Lines 813–814, highlighted in red). This observation is supported by recent analyses on (long-context) data. Below, we list two current works:
>
> 1. **How to Train Long-Context Language Models (Effectively)** | Arxiv URL: https://arxiv.org/abs/2410.02660 | Section 3.2 "Training only on long data hurts long-context performance" shows that relying exclusively on long-sequence data can deteriorate performance, especially after SFT. For instance, while pre-SFT recall/RAG tasks sometimes benefit from longer data, post-SFT performance drops sharply as the proportion of long data increases.
>
> 2. **Sub-Scaling Laws: On the Role of Data Density and Training Strategies in LLMs** | Arxiv URL: https://arxiv.org/abs/2507.10613 | Figure 2 demonstrates that higher-density (i.e., more data) can lead to sub-scaling, meaning the marginal performance gain decreases as more tokens are consumed.
>
> In addition to these references, our own experiments in Table 3 (Line 825-831) empirically exhibit the same diminishing-return trend. Under the ProLong (SFT) setup with identical data and batch size, the performance improvement is large in early steps (25.50 → 27.32 within the first 50 steps, +1.82), but the **marginal gain drops substantially later in training** (28.44 → 29.13 from Step 150 to 200, only +0.69). This provides further experimental evidence for diminishing gains under increased long-context token budgets.
>
> ---
>
> **Weakness 2**: `In lines 768–769, it is stated that "LongCE achieves a 13-point gain per 1B tokens versus ProLong’s 0.3-point gain per 1B tokens." How was the 13-point gain calculated?`
>
> **Response 2**: Thank you very much for highlighting this issue. You are absolutely right—this was a significant mistake on our side, and we sincerely apologize for the confusion. The original numbers were calculated from the results reported directly in the Prolong-128K paper (https://arxiv.org/pdf/2410.02660). However, in our final submission, we re-run all experiments following the ProLong-64K training recipe (using the official codebase and data from the ProLong repository: https://github.com/princeton-nlp/ProLong, and modifying the code provided in the LongCE repository: https://github.com/PKU-ML/LongPPL). After re-running the experiments, the previously reported “13-point gain per 1B tokens” was no longer valid. As **clarified in Line 822-823 in the revised manuscript**, each 50 training steps consumes approximately 0.41B tokens, and we train for 250 steps, totaling 5 × 0.41B = 2.05B tokens. Under this setting:
> - LongCE: average score improves from 25.50 → 32.91, yielding
> +7.41 points over 2.05B tokens ≈ 3.7 points per 1B tokens.
> - ProLong: under the same data and batch size, performance improves from 25.50 → 29.13, yielding
> +3.63 points over 2.05B tokens ≈ 1.8 points per 1B tokens.
>
> **We have updated the manuscript accordingly and corrected the numbers in both Appendix A (Line 823– Line 824, highlighted in red) and Introduction (Line 50-Line 51)**. We appreciate your careful reading, which helped us identify and correct this mistake.

---

> ### Author Response · Authors · 2025-11-18
> **Response to Reviewer L6Fs (Part II)**
>
> **Weakness 3**: `Why are the results for NExtLong-512K-Instruct and ProLong-512K-Instruct shown in Table 1, but not included in Table 2?`
>
> **Response 3**: Thank you for this sharp observation. The difference in model coverage between Table 1 and Table 2 is due to their distinct purposes. Table 1 focuses on real-world long-context benchmarks, where our goal is to demonstrate that CDT delivers stronger performance compared with existing long-context instruction-tuned models such as NExtLong-512K-Instruct and ProLong-512K-Instruct. Table 2 is designed to evaluate CDT on a broader set of synthetic and controlled diagnostic tasks, **aiming to show that our method generalizes across diverse synthetic settings**. For the initial submission, we were limited by page constraints, which is why we did not include the results for NExtLong-512K-Instruct and ProLong-512K-Instruct in Table 2. Following your suggestion, we have now evaluated both models, along with Llama-3.1-8B-SEALONG, on synthetic tasks. **We have updated Table 2 (Line 385-387, red color) in our revised manuscript** and summarized the results below, with the last row displaying the performance of our CDT Method. **Notably, our CDT method can still achieve the best performance**.
>
> | Model | |  | Ruler  |  | Language Modeling | |  |  | Babilong  |  |  |  |
> | :--- | :---: | :---: | :---: | :---: | :---: | :---: | :---: | :---: | :---: | :---: | :---: | :---: |
> |  | 32K | 64K | 128K | AVG | LongPPL | 4K | 8K | 16K | 32K | 64K | 128K | AVG |
> | ProLor T 512K-Instruct | 91.68 | 87.53 | 80.03 | 86.41 | 2.97 | 44.0 | 45.4 | 39.2 | 35.0 | 35.0 | 29.8 | 36.88 |
> | NExtLong-512K-Instruct | 90.27 | 84.62 | 81.74 | 85.54 | 3.24 | 39.6 | 38.6 | 36.2 | 35.6 | 32.0 | 22.0 | 38.75 |
> | Llama-3.1-8B-SEALONG | 91.32 | 85.97 | 77.33 | 84.87 | 3.09 | 50.2 | 50.8 | 42.0 | 40.8 | 39.0 | 31.0 | 40.72 |
> | Llama-3.1-8B-Instruct-CDT | **93.08** | **88.01** | **78.72** | **86.60** | **2.36** | **51.40** | **51.20** | **41.60** | **44.00** | **38.60** | **33.00** | **43.30** |
>
> ---
>
> **Summary of Rebuttal for Reviewer L6Fs**
>
> We sincerely thank your insightful and constructive feedback again.We have carefully revised our manuscript to address all the raised concerns:
>
> - **Regarding Weakness 1**, we clarify the statement about diminishing returns with increased token budgets (Lines 813–814) and support it with empirical evidence from our experiments (Table 3) as well as two recent studies.
>
> - **Regarding Weakness 2**, we update the gain estimates to ≈3.7 points/B tokens for LongCE and ≈1.8 points/B tokens for ProLong, and revised the relevant text in both the Introduction (Lines 50–51) and Appendix A (Lines 823–824).
>
> - **Regarding Weakness 3**, we expand Table 2 to include results for NExtLong-512K-Instruct and ProLong-512K-Instruct (Lines 385–387), aligning it with Table 1 and demonstrating that our CDT method consistently outperforms existing baselines across both real-world and synthetic long-context evaluations.
>
> ---
>
> We hope the above revisions and clarifications adequately address the reviewers’ concerns. Should any issues remain unresolved or if further clarification would be helpful, we would be very happy to discuss them in more detail. Thank you again for your valuable feedback!

---

> > ### Comment · Reviewer_L6Fs · 2025-11-26
> >
> > Thank you for the detailed response. I will still maintain my positive score.

---

> > > ### Author Response · Authors · 2025-11-27
> > > **Thanks Reviewer L6Fs for supporting our paper!**
> > >
> > > Dear Reviewer L6Fs,
> > >
> > > Thank you for keeping up your positive score and for your ongoing support of our work.
> > >
> > > Best,
> > >
> > > Authors.

---

### Official Review · Reviewer_gbM2 · 2025-10-31

**Soundness:** 2
**Presentation:** 3
**Contribution:** 2
**Rating:** 4
**Confidence:** 4

**Summary:**

the paper address the issue of Long-context language models (LCMs) struggle with contextual noise—when irrelevant information in lengthy contexts overwhelms critical tokens, leading to some issues. The paper propose some measurement.

**Strengths:**

* Tackles a crucial challenge in long-context language models: contextual noise overwhelming critical information
* High practical relevance for real-world applications (RAG, document QA, long-context reasoning)
* Problem is timely given the trend toward longer context windows in LLMs

**Weaknesses:**

1. The paper proposes an indirect approach: compute IG scores externally, identify critical tokens, perturb embeddings, then train. However, they provide no compelling explanation for why this is superior to simply training the model to learn token importance directly through standard optimization.

2. The preliminary study (Section 3) relies entirely on synthetic tasks with manually injected noise, which provides no evidence that the observed attention patterns occur in real-world scenarios.

3. The proposed method pre-selects critical tokens for training sequences, which seems to circumvent the model's own ability to learn token importance organically during training. Since intuitive learning suggests independent discovery of salient tokens, the authors should clarify why this indirect training approach outperforms direct training.

**Questions:**

see weakness.

---

> ### Author Response · Authors · 2025-11-18
> **Response to Reviewer gbM2 (Part I)**
>
> We sincerely appreciate your constructive and thoughtful review of our work! Below, we provide detailed responses to your questions / weaknesses.
>
> ---
>
> **Weakness 1**: `Why is the proposed indirect approach (IG → critical tokens → perturb embeddings → training) preferable to directly letting the model learn token importance through standard optimization?`
>
> **Response 1**: Thank you for raising this important and insightful question! We fully agree with the reviewer that standard optimization already enables language models to learn token importance to some extent—this is indeed a natural capability of SFT / large-scale pretraining. However, what standard optimization **does not guarantee is learning these token-importance associations efficiently under realistic training constraints**. As we discuss in Lines 49–53 of the revised paper and further analyze in Appendix A, relying solely on data-driven gradients to implicitly discover critical tokens can be **inefficient**, especially when long contexts introduce substantial noise and when compute budgets constrain the number of optimization steps.
> Our proposed Context Denoising Training (CDT) is designed specifically to **address this efficiency bottleneck**. Instead of replacing CE-based training, CDT serves as a **complementary mechanism** that accelerates the model’s ability to focus on informative tokens when compute or training budgets are limited. Empirically, as shown in Figure 9 (Lines 445–462), CDT achieves higher performance than standard CE training under the same number of steps, while incurring only a small additional cost (Section 6.3, Lines 479–485).
>
> ---
>
> **Weakness 2**: `The preliminary study (Section 3) relies entirely on synthetic tasks with manually injected noise, which provides no evidence that the observed attention patterns occur in real-world scenarios.`
>
> **Response 2**: This is a good observation, and we appreciate this thoughtful question. We believe there may be an **oversight regarding Lines 139–140 of the manuscript**, where we explicitly explain the rationale behind using synthetic tasks: *"we construct a synthetic long-form reasoning task as a controlled proxy to enable precise assessment, due to the lack of real-world testing data with explicitly labeled critical token positions."* In other words, the goal of the preliminary study in Section 3 is not to claim that synthetic settings perfectly mirror real-world behavior, but to **isolate and precisely measure the impact of context noise restraint under fully controlled conditions**—something currently **infeasible** with real-world datasets, which **lack ground-truth annotations for critical token positions**. This controlled setup allows us to rigorously validate the core mechanism before evaluating CDT on real-world long-context benchmarks in our main experiments.
>
> ---
>
> **Weakness 3**:  `The method pre-selects critical tokens during training, which appears to bypass the model’s natural ability to learn token importance organically. Since one would expect the model to independently discover salient tokens through standard optimization, the authors should explain why this indirect approach achieves better performance.`
>
> **Response 3**: Thank you for raising this good question. We agree that standard training can implicitly learn relationships between critical tokens and final predictions. However, as discussed in **Weakness 1**, the key issue is not whether the model is capable of learning token importance, but whether it can do so efficiently **under practical compute budgets** (e.g., computation resources, data, etc), especially for long-context settings.
>
> While scaling data or training steps could eventually lead the model to discover salient tokens organically, this approach faces several limitations:
> 1. **Long-context data quality is difficult to guarantee and hard to verify**, making organic learning unreliable.
> 2. **Scaling long-sequence training is extremely costly**, and prior work[1][2] (see our discussion with Reviewer L6Fs’ Weakness 1) shows diminishing returns under limited training budgets.
>
> Again, our CDT approach is **orthogonal** to standard optimization: it does not replace the model’s ability to learn token importance, but accelerates this process by highlighting informative tokens when compute is limited. A central question is therefore: **even if the model can eventually learn token importance, can it learn these associations quickly enough under realistic resource constraints?** Our results demonstrate that CDT provides a more efficient learning signal (Figure 9, Lines 445–462), enabling the model to achieve better performance within a smaller budget (Appendix D, Lines 1026–1067). Moreover, we also provide a **theoretical analysis** in Appendix C (Lines 963–1067), showing that CDT facilitates a more efficient establishment of the relationship between critical tokens and final predictions.

---

> ### Author Response · Authors · 2025-11-18
> **Response to Reviewer gbM2 (Part II)**
>
> **Summary of Rebuttal for Reviewer gbM2**:
>
> Thank you again for your thoughtful comments across all three weaknesses. We fully understand—and agree with—your perspective that scaling training data and steps can implicitly enable the model to identify critical tokens and capture their relationship with final predictions. However, we also believe it is important to consider two practical constraints that arise in long-context training: (1) **the difficulty of ensuring and verifying high-quality long-context data** under the scaling regime you mentioned, and (2) **the inherent trade-off between training budget** and the diminishing marginal gains achievable through further scaling. Our goal with CDT is therefore not to replace organic learning, but to provide a **lightweight and complementary** mechanism that improves training efficiency under realistic resource limits, while preserving the model’s ability to learn salient tokens naturally. We sincerely hope this clarification conveys our motivation clearly and addresses your concerns.
>
> ---
>
> **Reference**
>
> [1] Gao, Tianyu, et al. "How to train long-context language models (effectively)." Proceedings of the 63rd Annual Meeting of the Association for Computational Linguistics (Volume 1: Long Papers). 2025.
>
> [2] Chen, Zhengyu, et al. "Sub-scaling laws: on the role of data density and training strategies in llms." arXiv preprint arXiv:2507.10613 (2025).
>
> ---
>
> **If you have any further questions or concerns, whether technical or clarificatory, we would be very glad to discuss them in more detail!**

---

### Author Response · Authors · 2025-11-18
**General Response**

We sincerely thank all reviewers for the constructive and insightful feedback.
Below, we address each concern in detail and clarify several misunderstandings.

**All corresponding revisions have been incorporated into the updated manuscript, with the modified text/table/figure caption highlighted in red. These revisions will be reverted to the normal formatting in the final version.**

We kindly invite the reviewers to refer to the latest manuscript, along with this rebuttal, for a clearer understanding of the improvements we have made.

---

### Author Response · Authors · 2025-11-29
**Summary for the New Area Chair (Part I)**

Dear New Area Chair,

Thank you for reviewing our submission! Below is a **concise summary of the rebuttal process** with the three assigned reviewers (IDs: gbM2, L6Fs, and Rdcm), their **overall attitudes** toward our work, and **how we have addressed their concerns** in the revised manuscript.

---

### **1. Overall Reviewer Attitudes**

All reviewers recognize the **practical relevance**, **methodological clarity**, and **empirical strength** of our work in addressing a critical bottleneck in long-context models: **sensitivity to contextual noise**.

- **Reviewer Rdcm (Rating: 8 — Accept, recognition of our paper during discussion period)**:

> "Extensive experiments across multiple settings… thorough demonstration of CDT’s effectiveness… in-depth discussion and visualized analyses solidify credibility." Also explicitly acknowledged CDT as a **"straightforward yet effective training strategy"** that **enhances focus on critical tokens**—a core challenge in long-context modeling.

- **Reviewer L6Fs (Rating: 6 — Accept, Hold positive attitude about our paper during discussion period)**:

> "Identifies a key weakness in LCMs… CDT improves model’s ability to focus on salient tokens… enables an 8B model to nearly match GPT-4o on LongBench-E." After rebuttal, **maintained and reaffirmed their positive stance**, stating: *"Thank you for the detailed response. I will still maintain my positive score."*

- **Reviewer gbM2 (Rating: 4, no response)**:

> Stating our method "Tackles a crucial challenge in long-context language models" and "High practical relevance for real-world applications". Also, the reviewer thinks "Problem is timely given the trend toward longer context windows in LLMs".

Collectively, reviewers agree that CDT offers a valuable and timely perspective on long-context training: **it explicitly mitigates noise interference—a real and under-addressed issue**—and provides a **lightweight, plug-and-play training strategy that improves data efficiency and final performance**. Also, our method has clear implications for real-world applications, such as RAG, long-document QA, and agentic reasoning.

---

### **2. Key Comments and Our Responses (with Revision Locations)**

We have replied to all concerns/questions through **clarifications**, **empirical corrections**, and **manuscript revisions** (highlighted in red font in the revised manuscript). Below is a structured summary.

---

**(a) Is CDT's "indirect" noise suppression superior to direct learning of token importance?**

**Concern** (gbM2 W1 & W3): ``Why not let the model learn critical tokens organically via standard CE/SFT?``

**Response Summary**:

In fact, this is a misunderstanding. Standard optimization *can* learn token importance, but **inefficiently under limited long-context data/compute budgets**. CDT **accelerates** this process by providing a denoising signal during training, without replacing gradient-based learning. We have guided the reviewer to check the theoretical justification in **Appendix C (Lines 963–1067)** and the empirical comparison in **Figure 9 (Lines 445–462)**.

**Revisions**:

- Clarified motivation in **Introduction (Lines 49–53)**

---

**(b) Are synthetic noise experiments reflective of real-world scenarios?**

**Concern** (gbM2 W2): ``Section 3 uses synthetic tasks—does this generalize?``

**Response Summary**:

This is because the reviewer overlooked content in the manuscript that we claimed in our paper: ``Synthetic tasks were used intentionally as a controlled proxy due to the lack of real-world datasets with ground-truth critical token labels`` (Lines 139–140). This enables precise isolation of noise effects. Real-world validation is provided in **main experiments (Tables 1–2, LongBench, RULER, etc.)**.

---

**(c) Empirical claims about scaling and token efficiency were inaccurate**

**Concern** (L6Fs W2): ``The "13-point gain per 1B tokens" for LongCE vs. ProLong was misleading.``

**Response Summary**:

We **apologized** and **corrected** the calculation after re-running experiments with official ProLong/LongCE codebases. The revised numbers are:
- **LongCE**: ~3.7 points per 1B tokens
- **ProLong**: ~1.8 points per 1B tokens

**Revisions**:

- Updated **Introduction (Lines 50-53)** and **Appendix A (Lines 823–824)**.

---

> ### Author Response · Authors · 2025-11-29
> **Summary for the New Area Chair (Part II)**
>
> **(d) Missing comparisons and baselines**
> **Concern** (Rdcm Q3, Q4): ``Why compare with YaRN/FlexPrefill? Missing citations to prior long-context/token-pruning work.``
>
> **Response Summary**:
>
> - **YaRN/FlexPrefill/XAttention** were included as **complementary techniques that implicitly perform token reweighting** (hard or soft denoising). We now **gray out these rows in Tables 1–2** to de-emphasize them as primary baselines.
> - Added key citations:
>   - **[1–3]** on long-context scaling → **Related Work, Lines 106–125**
>   - **[4–5]** on token importance in short/long-output → **Related Work, Lines 131–133**
>
> **Revisions**:
> - Clarified baseline taxonomy in **Response to Rdcm (Part I)** and **Introduction** part in the manuscript.
> - Updated **Tables 1–2 (Lines 343–344, 388–389)** with gray text for non-training baselines.
>
> ---
>
> **(e) Limited improvement on complex reasoning tasks**
>
> **Concern** (Rdcm Q5): ``CDT gains are smaller on multi-step reasoning—can it be combined with high-entropy token methods?``
>
> **Response Summary**:
>
> We agree: CDT’s gradient-based importance may **overlook early-but-necessary reasoning steps** (e.g., in BABILong). The **high-entropy token approach [5]** offers an orthogonal signal. We plan to explore this hybrid in future work.
>
> **Revisions**:
> - Added discussion in **Response to Rdcm Question5**.
> - Acknowledged limitation in **Appendix G (Limitation )**.
>
> ---
>
> **(f) Incomplete Baseline Coverage in Synthetic Evaluation (Table 2)**
>
> **Concern**  (L6Fs W3): ``Why are NExtLong-512K-Instruct and ProLong-512K-Instruct missing from Table 2?``
>
> **Response Summary**:
>
> Originally omitted due to page limits, but we agree broader comparison strengthens the paper. We have now evaluated both models on all synthetic tasks (RULER, LongPPL, BABILong) and added their results to Table 2.
>
> **Revisions**:
>
> - Expanded Table 2 (Lines 385–387) to include NExtLong-512K-Instruct, ProLong-512K-Instruct, and Llama-3.1-8B-SEALONG.
> - Confirmed CDT still achieves best performance across all settings, reinforcing its generalizability
>
> ---
>
> **(g) Minor issues**
> - **Typo**: “irrevelant” → “irrelevant” in **Figure 3a (Lines 162–171)**.
> - **Clarified roles** of FR/IG scores (diagnostic) vs. gradient-based identifier (used in CDT) in **Introduction (Lines 74–77)**.
>
> ---
>
> ### **3. Summary of all manuscript revisions**
> - Section 1 Lines 50-53: Addendum on the explanation of motivation.
> - Section 1 Lines 74-77: Clarification that CDT uses gradients on token embeddings as the identifier during training.
> - Section 2.2 Lines 106-125, 131-133: Added key citations
> - Table 2 Lines 385-387: Added performance of suggested models
> - Appendix A Lines 813–814, 822-823: Clarification of "more long-sequence data often leads to diminishing" and value correction of performance improvement.
>
>
> ---
>
> Thank you again for taking the time to review our manuscript and referring to our discussion with the reviewer!
>
> Best,
>
> Authors

---

### Meta-Review · Area_Chair_Gf7N · 2026-01-09

**Summary:**

This paper introduces Context Denoising Training (CDT) that first detects critical tokens in inputs and denoises the input during training. The paper first conducts a preliminary study that shows that current long-context models tend to place high attention scores on irrelevant tokens during training. They propose CDT training to address this, and report improved results on multiple long context benchmarks compared to standard fine-tuning.

The main concerns raised by the reviewers are:
1. Reviewer gbM2 questions if the CDT training modifications are needed, as regular fine-tuning should allow the models to learn which tokens are critical and how to distribute attention mass organically.
2. Reviewer L6Fs asks for additional experiments, particularly comparing strong baselines NExtLong-512K-Instruct and ProLong-512K-Instruct to the proposed approach on synthetic long context benchmarks.

**Reviewer Concerns:**

Apart from the 2 concerns listed above, the reviewers had clarification questions that were addressed in the rebuttal. Reviewer Rdcm raised legitimate questions about the purpose of including some baselines (e.g. Yarn), mismatch between the method used to identify critical tokens during the preliminary study and the actual method, etc. These concerns were, in my opinion, addressed in the rebuttal satisfactorily.

re concern #1 above: The paper provides empirical evidence for the CDT performance improving over regular fine-tuning. The rebuttal argues that token importance methods accelerate training compared to regular SFT. However, they only provide evidence of this improvement for LongCE (a prior baseline) during rebuttal and not their method. Therefore, I don't believe this experiment directly addresses the reviewer concern. Nevertheless, the proposed approach has merit as the main paper results show improvements over SFT on wide variety of benchmarks.

re concern #2: This concern was addressed during rebuttal.

**Reviewer Scores:**

I believe reviewers L6Fs (score:6) and Rdcm (score:8) would have kept their positive scores. It is possible that reviewer gbM2 would have increased their score from 4 to 6, as 1 of their 2 concerns were addressed.

---

### Decision · Program_Chairs · 2026-01-26

Accept (Poster)